# On the social dynamics of moisture recycling

Patrick W. Keys[1,2] and Lan Wang-Erlandsson[2,3]

[1]School of Global Environmental Sustainability, Colorado State University, Fort Collins, USA
[2]Stockholm Resilience Centre, Stockholm University, Sweden
[3]Research Institute for Humanity and Nature, Kyoto, Japan

*Correspondence to:* patrick.keys@colostate.edu

**Abstract.** The biophysical phenomenon of terrestrial moisture recycling connects distant regions via the atmospheric branch of the water cycle. This process, whereby the land surface mediates evaporation to the atmosphere and the precipitation that falls downwind, is increasingly well-understood. However, recent studies highlight a need to consider an important and oft missing dimension - the social. Here, we explore the social dynamics of three case study countries with strong terrestrial moisture recycling: Mongolia, Niger, and Bolivia. We first use the WAM-2layers moisture tracking scheme, and ERA-Interim climate reanalysis, to calculate the evaporation sources for each country's precipitation, aka the precipitationshed. Second, we examine the social aspects of source and sink regions, using economic, food security, and land-use data. Third, we perform a literature review of relevant economic links, land-use policies, and land-use change for each country and its evaporation sources. The moisture recycling analysis reveals that Mongolia, Niger, and Bolivia recycle 13%, 9%, and 18% of their own moisture, respectively. Our analysis of social aspects suggests considerable heterogeneity in the social characteristics within each country relative to the societies in its corresponding evaporation sources. We synthesize our case studies, and develop a set of three system archetypes that capture the core features of the moisture recycling social-ecological systems (MRSES): isolated, regional, and tele-coupled. Our key results are: (a) geophysical tele-connections of atmospheric moisture are complemented by social tele-couplings forming feedback loops, and consequently, complex adaptive systems; (b) the heterogeneity of the social dynamics among our case studies renders broad generalization difficult, and highlights the need for nuanced individual analysis; and, (c) there does not appear to be a single desirable or undesirable MRSES, with each archetype associated with benefits and disadvantages. This exploration of the social dimensions of moisture recycling is part of an extension of the emerging discipline of socio-hydrology, and a suggestion for further exploration of new disciplines such as socio-meteorology or socio-climatology, within which the Earth system is considered as a co-evolutionary social-ecological system.

## 1 Introduction

Humanity is unequivocally leaving its mark on Earth, in terms of changes to the land surface (Ellis and Ramankutty, 2008), biostratigraphic layers (Zalasiewicz et al., 2015), and global hydrologic cycles (Zhou et al., 2016). Against this backdrop of heedless, anthropogenically-driven Earth system change, there have emerged new insights into the interactions between land-use change and the atmospheric branch of the water cycle. That land-use change can and does influence the atmospheric water cycle is well-supported (e.g. Lo and Famiglietti, 2013; Wei et al., 2013; Halder et al., 2016; de Vrese et al., 2016). Impacts can

include modifications of the energy budget (e.g. Swann et al., 2015), impacts to local or regional circulation (e.g. Tuinenburg et al., 2013), and impacts to the atmospheric water cycle (e.g. Spracklen et al., 2015; Badger and Dirmeyer, 2015).

The general process of water evaporating from the surface of the Earth, traveling through the atmosphere as water vapor, and eventually falling out as precipitation downwind is known as moisture recycling (Lettau et al., 1979; Koster et al., 1986). The component of this process that takes place over land is often distinguished as terrestrial moisture recycling (as opposed to oceanic moisture recycling) (van der Ent et al., 2010). However, in the context of this paper and for the sake of brevity, we will use the phrase moisture recycling to refer specifically to the terrestrial component. Considering the recent debate regarding human impacts to large-scale hydrology (Rockström et al., 2012; Heistermann, 2017), there is a need to clarify and highlight the importance of anthropogenic modification of terrestrial moisture recycling. Moreover, it is incumbent on the scientific community to begin unpacking how the constituent components of the Earth system interact with the *social*.

Any research on moisture recycling that is either driven by anthropogenic land-use change or is seeking to understand how changing moisture recycling impacts land-use, has a social focus. The range of social topics that have been explored in the context of moisture recycling include: natural hazards and flooding (Dirmeyer and Brubaker, 1999; Dominguez et al., 2006), irrigation impacts to moisture recycling (Lo and Famiglietti, 2013; Tuinenburg et al., 2014), rainfed crop production (Bagley et al., 2012; Keys et al., 2012), ecosystem services (Ellison et al., 2012; Keys et al., 2016), urban water vulnerability (Keys et al., 2018), and even the import and export of moisture among nations (Dirmeyer, P. A. et al., 2009; Keys et al., 2017). Land-use change impacts to rainfall via moisture recycling has been suggested to be potentially linked to the Sahel drought during the 1970s and this moisture recycling mechanism was suggested to be an integral part of the socio-hydrology discipline (Sivapalan et al., 2012). Socio-hydrology has expanded to include many social dynamics, including explicit inclusion of systems-thinking (Pande and Sivapalan, 2015), yet integration of the social dynamics of moisture recycling remain largely unexploredThese preliminary examinations of moisture recycling in the context of social issues have been revealing, but the social domain has often been investigated outside the biophysical system feedback boundaries, either as initiator of hydrological change (e.g., land-use change) or as receiver of hydrological change (e.g., decreasing crop yields), but almost never as dynamic modifier of hydrological change.

Social-ecology, however, departs from the view that human and environmental systems are separate, and that social-ecological systems (SES) are tightly coupled complex adaptive systems. In its simplest form, the classic social-ecological systems diagram (Holling, 2001; Folke et al., 2004; Folke, 2006) includes an ecological node, a social node, and arrows connecting the two nodes to one another, as well as feeding back on themselves. Recognizing that systems exist within hierarchies, the concept of panarchy was developed for understanding the cyclic dynamics of combined human-nature systems across scales (Gunderson and Holling, 2002). For the purpose of analysing the likelihood of self-organisation in SES, Ostrom (2009) proposed a general multilevel, nested framework comprised of resource system, resource unit, governance system, and users, which are all embedded within ecological as well as social-economical-political settings. In terms of water management and governance, this type of SES thinking have been primarily applied to rivers and lakes (Cosens and Williams, 2012; Gunderson et al., 2017), but not to atmospheric moisture flows.

## 1.1 Research questions and rationale

Here, we take inspiration from the SES type of thinking to address the social dynamics of moisture recycling by posing the following questions:

1. How are moisture recycling patterns interlinked with social dynamics?

2. Are there dynamic social connections that link precipitation sinks and sources?

3. What are the system architectures that create feedbacks among geophysical, ecological and social drivers?

We want to be clear that our analysis includes both objective analysis of moisture recycling and social data, as well as subjective assessment of ongoing policy and management activities related to land-use change. This blending of quantitative and qualitative, as well as objective and subjective, is at the heart of our approach to understanding social dynamics of moisture recycling.

This work will be useful for three key reasons. First, the conceptual approach will provide Earth system scientists generally, and hydrologists specifically, with the basics of how social systems (that are the sinks of upwind moisture recycling) are connected in many different ways back to the moisture sources. Second, this conceptual insight provides an entry-point for more accurate modeling of the feedbacks that could affect moisture recycling patterns (rather than only considering, e.g. geophysical phenomena). Third, this manuscript provides insights for resource managers, particularly land and water managers, who are searching for new leverage points within their dynamic social-ecological systems. Understanding where key feedbacks, bottlenecks, and potential cascades are located within a system can provide managers with better information about the consequences of direct or indirect intervention within their systems.

We argue that exploring the social dynamics of moisture recycling improves our understanding Earth system dynamics by providing general insight into how humanity modifies the Earth system, but also into the heterogeneity of moisture recycling social-ecological systems.

## 2 Methods and Data

## 2.1 Conceptual development

We propose to develop a framework for moisture recycling social-ecological systems (SES), starting with the classical social-ecological systems concepts (Holling, 2001; Folke et al., 2004; Folke, 2006) (Fig 1a). A key feature of SES are the connections among, different system components, creating loops. These system loops allow for feedbacks within the system, leading to complex, emergent behavior. Likewise, the systems can adapt to changing conditions, making them complex adaptive systems. In parallel, we use the simplest representation of a moisture recycling system, which comprise of the source of evaporation and the sink of precipitation (Fig 1b). Typically, the direction flows from the source to the sink, with some amount of internal recycling within the source as well as the sink. The absence of a connection from the sink back to the source is because

the moisture recycling relationship is based on purely biogeophysical connections, with water evaporating from the source, traveling along prevailing wind currents as water vapor, and condensing and falling as precipitation elsewhere. However, it is possible to place an SES within both the source and sinks (Fig 1c). This nesting of SES within the moisture recycling system suggests the emergence of a social connection back from the evaporation source to precipitation sink (Fig 1c, red), and represents the fundamental basis for our current research. Furthermore, this concept of a Moisture Recycling Social-Ecological System (MRSES) will be used to develop archetypes that can guide model development and practitioner prioritization. To construct the archetypes, we explore three case studies to identify how the geophysical and social connections interact to create feedbacks, and how these feedbacks lead to broader system dynamics. A table of key terms is provided for reference, especially given the interdisciplinary nature of this research (Tab 1).

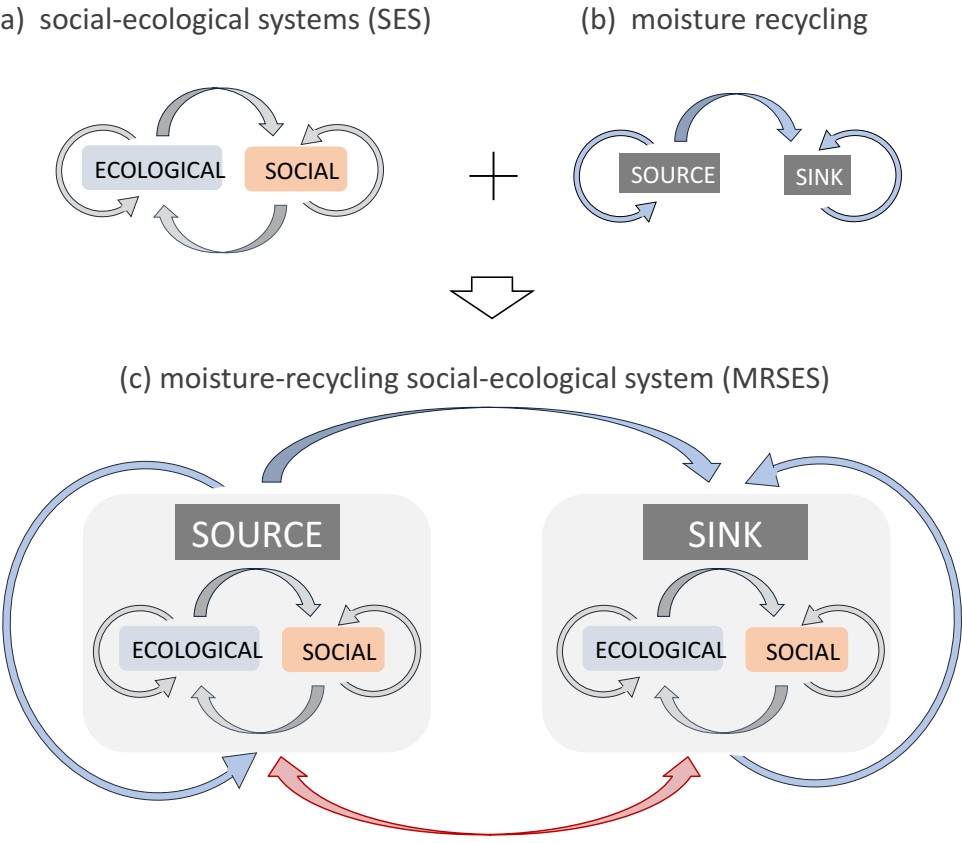

**Figure 1.** Conceptual construction of moisture recycling social ecological system (MRSES) archetypes. A hypothetical, idealized social-ecological system (a) is nested within the idealized moisture recycling system (b), creating a linked moisture recycling social-ecological system (c). The red arrow at the bottom of (c) emphasizes the fact that the social aspects of a MRSES are what link the precipitation sink back to its evaporation sources. The arrow points in both directions, since social links are not restricted by biophysical constraints.

**Table 1.** Definitions of key terms used in this research.

| KEY TERM | DEFINITION |
| --- | --- |
| Moisture recycling | The process of evaporation arising from the surface of the Earth traveling through the atmosphere and returning elsewhere as precipitation. |
| Precipitationshed | An area B that is upwind of region A, that provides evaporation for A's precipitation. The extent of precipitationshed is determined by the area that contributes moisture (source region) to a selected region (sink region). |
| Source region | The source of evaporation (in a moisture recycling context) |
| Sink region | The sink of precipitation (in a moisture recycling context). A user-selected region for which the source of precipitation is tracked and determined. |
| Complex adaptive systems | A system with many parts that interact to produce system-wide behavior that cannot easily be explained in terms of interactions between the individual constituent elements. |
| Social Dynamics | Social drivers, feedbacks, interactions or other social features of a complex adaptive system. |
| Social-ecological systems | A system in which human and nature are intertwined, and system dynamics include both biophysical and social feedbacks. |
| MRSES | Moisture recycling social-ecological system. A framework for considering social, ecological, and moisture recycling feedbacks. |
| Tele-connection | A connection between bio- or geophysical processes that are separated by time, space, or both. |
| Tele-coupling | A connection among social processes that are separated by time, space, or both. |
| Socio-hydrology | The emerging scientific discipline that jointly considers hydrological and social dynamics. |

## 2.2 Case study selection

To find regions that are both relevant to terrestrial moisture recycling dynamics, as well as (potentially) relevant to social dynamics, we use the regions selected from the work in (Keys et al., 2016), which identified global regions that are particularly reliant on current vegetation for rainfall from moisture recycling (Keys et al., 2016, see Fig 2b). The regions that receive the most (relative) precipitation from upwind vegetation-regulated evaporation are East and Central Asia, parts of the Sahel, southwestern Africa, the southern Amazon and La Plata river basin in South America, and much of Canada. We also aimed to select regions that were 'socially-relevant' which, in this work, we define as having potential social-ecological inequalities (e.g. differences in child malnutrition) among the downwind beneficiaries of moisture recycling and the upwind providers of moisture recycling. Furthermore, since this research is about social dynamics, we select countries as our unit of analysis, rather than hydrological units (e.g. basins) or biophysical units (e.g. specific landscapes).

Based on these criteria, the selected case studies are Mongolia, Niger, and Bolivia. These three sink regions are distributed across three continents, providing separate moisture recycling dynamics, and distinct social systems, while having a comparable similar spatial footprint (with subsequently comparable moisture recycling source and sink footprints).

For each of these case studies, we identify: (a) the discrete sources of evaporation falling as precipitation within the case study, i.e. the precipitationshed (Keys et al., 2012); (b) the dominant land-use types that are present in the sink and the precipitationshed; (c) a quantitative comparison of social dynamics in sources and sinks; and, (d) a qualitative literature review of the types of social dynamics present within the precipitationshed.

## 2.3 Tracking the sources of moisture

We use an 'offline Eulerian' moisture tracking scheme called the Water Accounting Model-2layers, hereafter, WAM-2layers (for original model configuration, van der Ent et al. (2010); for two-level update van der Ent et al. (2013)). For a single gridcell and corresponding column of air, the model works as follows: first, the amount of moisture entering the column as evaporation is tracked; second, the evaporated water mixes with the moisture in the lower and upper levels of the column; third, moisture blows into and out of the column from adjacent columns; and, fourth, a certain amount of precipitation exits the lower level of the column. This tracking procedure is replicated across the entire planet for each timestep of the model. In this way, moisture can be tracked across the entire planet, simultaneously.

We use the backtracking feature of the WAM-2layers (Keys et al., 2012), which allows for the identification of the source region of precipitation - i.e., the precipitationshed. As input to the WAM-2layers, we use ERA-Interim Reanalysis data, from the European Center for Mesoscale Weather Forecasting (Dee et al., 2011). We downloaded global, model-level data at the 1.5 deg. by 1.5 deg. resolution. The WAM-2layers uses 6-hourly data for horizontal and vertical wind, humidity, and surface pressure; and it uses 3-hourly data for evaporation and precipitation. The data are interpolated into two levels, an upper- and lower-level of the atmosphere, to accommodate the upper and lower atmosphere processes, namely wind shear (van der Ent et al., 2013), and this separation roughly corresponds to the 800 mb level. We emphasize that the WAM-2layers is a moisture tracking scheme, and not a simulation. It is possible to couple the WAM-2layers with dynamic simulations of land-surface

hydrology, including vegetation (Wang-Erlandsson et al., 2016; Keys et al., 2016), but that is not what we have done in this research. Thus, the results that we present are purely based on the implicit hydrological information contained within the ERA-Interim Reanalysis data.

Many approaches for precipitationshed boundary selection have been described (Keys et al., 2012, 2014, 2017), and we will employ the 1mm boundary as used in Keys et al. (2014). The 1mm boundary refers to a boundary that includes all regions that contribute 1mm or more of annual precipitation to the sink region. The precipitationshed is the spatial footprint that we will use in our analysis of the social dynamics among the precipitation sink and its sources of evaporation.

## 2.4   Quantifying social features of the precipitationshed

One approach to capturing the social dynamics of moisture recycling is to characterize various social, economic, and other factors that are biophysically-relevant and that can provide insight into the dynamics among the sources and sinks of moisture. In our analysis, we use land-use based on anthropogenic biomes data, i.e. anthromes (Ellis and Ramankutty, 2008), food security using the proxy of child malnutrition (Socioeconomic and Data Applications Center SEDAC, 2005), and economic wealth using the proxy of market influence Verburg et al. (2011) (Table 2). The anthromes data are a land-use classification scheme explicitly developed to account for both the various human uses of landscapes and density of human populations. For example, rather than having land-use categories that are uniformly 'cropland', the anthromes data range from densely populated 'Rice villages' (>100 people per $km^2$), to sparsely inhabited 'Remote croplands' (<1 person per $km^2$).

The child malnutrition data represent the number of malnourished children per thousand under the age of five years, and is available at the scale of countries (e.g. Russia has a single value), as well as sub-national (e.g. Sudan has may different values). The market influence data is a calculated from a variety of other datasets, and is a combination of (1) access to markets (calculated using data on infrastructure, travel distance, and travel costs to major cities), and (2) per capita GDP (for more on calculation, see (Verburg et al., 2011)). As seen in Table 2, these datasets are spatially-gridded, at various resolutions. We interpolated the various data to the resolution of the moisture recycling information (i.e. 1.5 deg. by 1.5 deg.) so that they were comparable with one another in subsequent analyses of the sources and sinks of moisture.

Note, the 'source' results (presented in the social characteristics figures, Figs 2-4), refer to the sources of precipitation excluding the sink itself. In other words, despite the presence of internal moisture recycling, the description of source characteristics do not include the case study country itself.

## 2.5   Literature-review of social dynamics

To complement the quantitative characterization of the precipitationsheds, we performed a literature review focused on each of the case study regions (i.e. Mongolia, Nigeria, and Bolivia), exploring potential dynamics that exist among the social, biophysical, and other aspects of the precipitationshed. The literature review is specifically intended to help reveal some of the qualitative, social interactions (e.g. land use policies, regulatory interactions, economic interlinkages) that quantitative analysis cannot provide. We used the hypothetical moisture recycling social-ecological system (MRSES) concept diagram as a guiding heuristic for how to search for important dynamics.

**Table 2.** Summary of metadata for anthromes, child malnutrition, and market influence data.

| VARIABLE NAME | DESCRIPTION | SOURCE RESOLUTION | UNITS | TIME PERIOD | SOURCE |
|---|---|---|---|---|---|
| Anthromes | Anthropogenic biomes, incl. type of human land-use and population density | 5 arc minute | [categories] | 2000 to 2005 | Ellis and Ramankutty, 2008 |
| Child malnutrition | The number of children under five years of age, who are malnourished, per 1000 children | sub-national or national | #/1000 | 2005 | SEDAC, Columbia U, |
| Market influence | Per capita GDP multiplied by market access (e.g. composite of travel distance, time, etc) | 5 arc minute | $/ person | 2010 (per cap GDP), 1979-2005 (market access) | Verburg et al., 2011 |

For each case study, the general approach was to use the precipitationshed as the spatial boundary within which we evaluated the dominant processes governing land-use change and the types of dynamics among the sink region and the source regions. A blend of journal articles, grey literature, and web sources provided the key information for building the qualitative description of these social dynamics. The result of the case study analysis is a blend of quantitative and qualitative information, which combined to form a coherent representation of the social dynamics of moisture recycling for each case study.

## 2.6 Construction of archetypes

System dynamics models expose how different components of a system interact with one another, and they can help reveal the relative importance of different connections and interactions. We distilled the insights from the three case studies into the creation of several MRSES archetypes. The conceptual model presented in Fig 1 formed the basis of the MRSES, and was infused and modified using the case studies.

## 3 Results

### 3.1 Mongolia case study

#### 3.1.1 Precipitationshed analysis

In Mongolia, the precipitation source is located in the northern half of the country and along its north-western border (Fig 2a). Mongolia provides about 13% of its own precipitation largely from the northwestern half of the country. Aggregated over space (Fig 2b), the largest moisture contributor of Mongolia's precipitation is Russia (29%), primarily due to the very large spatial extent of Russia. The only large evaporation contributions originate in the Russian steppe between Kazakhstan and Lake Baikal, and the rest are very small amounts across much of the Russian land surface. The same is generally true of China

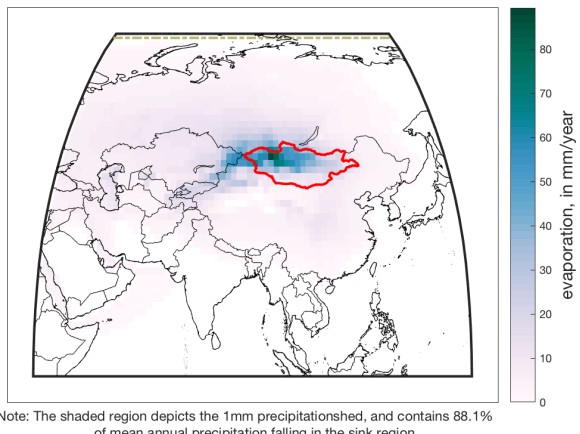

(a) Mongolia precipitationshed

Note: The shaded region depicts the 1mm precipitationshed, and contains 88.1% of mean annual precipitation falling in the sink region.

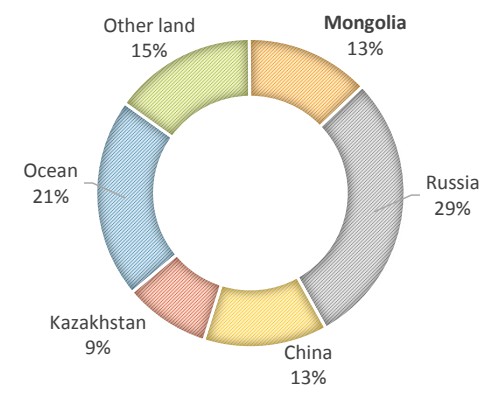

(b) Origin of precipitation by nations

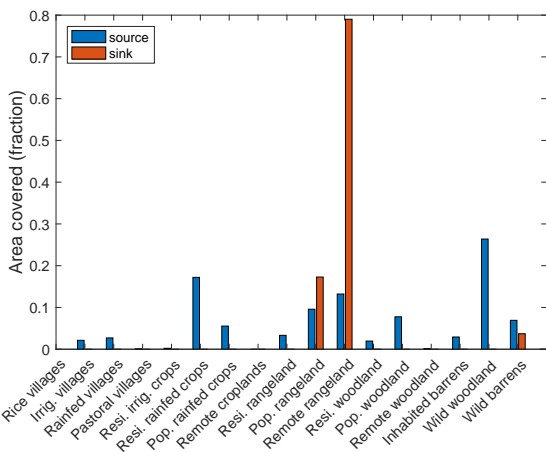

(c) Distribution of anthromes in Mongolia case study

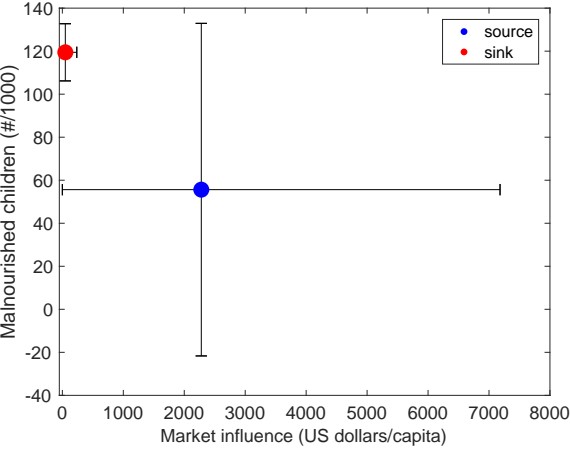

(d) Hunger and wealth in Mongolia case study

**Figure 2.** Mongolia case study including (a) precipitationshed analysis, (b) key source countries, (c) comparison of land-uses between country and precipitationsheds, and (d) comparison of economic and food security aspects of country and source regions.

(13%), with significant contributions from the Tian Shan mountains to Mongolia's west, and small contributions from the rest of China's land surface. The eastern tip of Kazakhstan is also a key evaporation source region (9%), with smaller contributions from the rest of central Asia.

### 3.1.2 Social characteristics

Mongolia is predominantly classified as remote rangeland ( 80%) with some populated rangelands and wild barren land (Fig 2c). Mongolian society is generally quite poor (mean market influence of $48.40 per capita), with relatively high child malnutrition (around 12% or 120 in 1000 kids) (Fig 2d). Most of Mongolia's evaporation key sources are rangelands (within and outside Mongolia), as well as wild and remote woodlands. The societies in Mongolia's source regions are on average richer (market influence of $2,279.50), and experiencing less hunger (5.6%).

### 3.1.3 Literature-review of social dynamics

Mongolia's precipitationshed includes significant contributions from internal recycling within Mongolia, as well as significant contributions from the East Siberian Taiga to the north, the steppes in Kazakhstan, and Xinjiang province in northwestern China. Mongolia's land-use policy has had several distinct phases of management in the recent past, with traditional management of grasslands via customary nomadic herder institutions, then with the *negdel* 'pastoral cooperative' policy, followed by post-*negdel* policies largely dependent on local government management (Ojima and Chuluun, 2008). The transition from widespread mobility of herders to much more confined mobility (in large part due to expansion of agricultural lands) has led to significant changes in land- and water-use. Recent analyses find that if present trends of agricultural expansion continue, then water shortages may become common (Priess et al., 2011). Similarly, if irrigated agricultural continues for a significant period of time, and soils are not drained properly (as has happened throughout much of Inner Mongolia in China), then it is possible that soils and landscapes will become salinated and less able to sustain vegetation (Kendy et al., 2003). The segmentation of Mongolia's traditionally managed grassland landscapes into grazing land and agricultural land, and the associated fragmentation or prohibition of seasonal movements of livestock, may lead to significant land-use change in the near future.

Kazakhstan, to Mongolia's west, provides a significant amount of moisture especially from its northern steppes, and from the Altai and Tien Shan mountains. Following the collapse of the Soviet Union, Kazakhstan's livestock population decreased significantly, with a concurrent reduction in grazing land pressure (Robinson et al., 2003). This change in grazing pressure has led to replacing of previously grazed land with other grasses and weedy species during fallow periods, and importantly has led to detectable changes in vegetation and associated changes in near-surface meteorology (De Beurs and Henebry, 2004). This has direct implications for evaporation and subsequently moisture recycling.

In terms of politics, the ascendent Mongolian People's Party has strong ties to Russia's Vladimir Putin, suggesting that political and diplomatic levers of power may flow not through adjacent China, but rather north to Russia (Jargalsaikhan, 2017). Likewise, Mongolia is currently experiencing significant crises with regard to its management of debt, and thus it is beholden to both international lending agencies, as well as the international mining conglomerates fueling its development. Though Mongolia is reliant on China for export goods (China is the recipient of 79% of Mongolia's trade by volume), China is not reliant on Mongolia in nearly the same way (Simoes and Hidalgo, 2011). Likewise, Mongolia relies on imports from China and Russia in nearly equal measure (31% and 26%, respectively) (Simoes and Hidalgo, 2011). Aside from mining concessions and the associated resource use, these political and economic connections are not directly linked to large-scale land-use change,

but rather to the underlying conditions and connections that might provide motivation (or lack thereof) for managing land-use in a manner that is most sustainable for moisture recycling specifically, and water resources generally.

To summarize, Kazakhstan's abandonment of former grazing land, the low level of land-use change in the East Siberian Taiga, and the isolation of the Altai and Tien Shan mountain regions suggest relatively low social connectivity from source to sink. Likewise, the fact that Mongolian institutions are stronger (than e.g. in Kazakhstan and in Xinjiang) and that the departure from historic land-use and land-management is more pronounced in Mongolian grasslands, we suggest that the social connections are strongest within Mongolia itself, leading to a somewhat isolated state of precipitationshed social connectivity.

## 3.2 Niger case study

### 3.2.1 Precipitationshed analysis

In Niger, the precipitation source is concentrated along the southern border (Fig. 3), with a moisture supply plume fading towards the south. Niger supplies just 9% of its own moisture, but because of the large number of neighboring countries, this percentage suffices to make Niger top the list over other moisture suppliers. There is significant contribution from Nigeria to the South (6%), and Chad (6%) and Sudan (5%) to the east. There is some contribution from coming from the Democratic Republic of the Congo, the Central African Republic, and South Sudan (all 2%). Beyond that, there is diffuse contribution from some West African countries, as well as the Mediterranean.

### 3.2.2 Social characteristics

Land-use in Niger is predominantly "wild barrens", given that much of its land surface is in the Sahara. Among the populated anthromes, Niger is equally distributed among rainfed croplands, rangelands, and inhabited (i.e. very low population density) barren lands. Residential rainfed crops are a dominant anthrome type among Niger's moisture sources, along with some range-lands and woodlands. Niger's hunger and wealth characteristics are different from its source countries with the mean values of $49.20 per capita market influence, and 40.2% child malnutrition, barely within the standard deviation of the societies in its evaporation sources. The average market influence among Niger's sources is $4,081, with a very wide standard deviation ranging from nearly $0 to more than $11,000. Likewise, the mean child malnutrition is 23.8% with a standard deviation from a high of 40% to a low of around 7.5%.

### 3.2.3 Literature-review of social dynamics

Niger generates a significant fraction of its own rainfall, primarily from the southern section of the country that is used for semi-arid agriculture and grazing. Land use in Niger is varied, ranging from barren deserts in the north, to livestock grazing and rainfed agriculture in the south. Land-use change over the last several decades has seen increases in cropland cover (where possible) with corresponding decreases in fallow land (Hiernaux et al., 2009). The ownership of land, i.e. land tenure, in Niger has historically been governed by customary systems administered by communal Chiefs. However, in the mid-1980s there was a push to formalize land tenure via government-sponsored registration efforts, especially in rural areas of Niger (Toulmin, 2009).

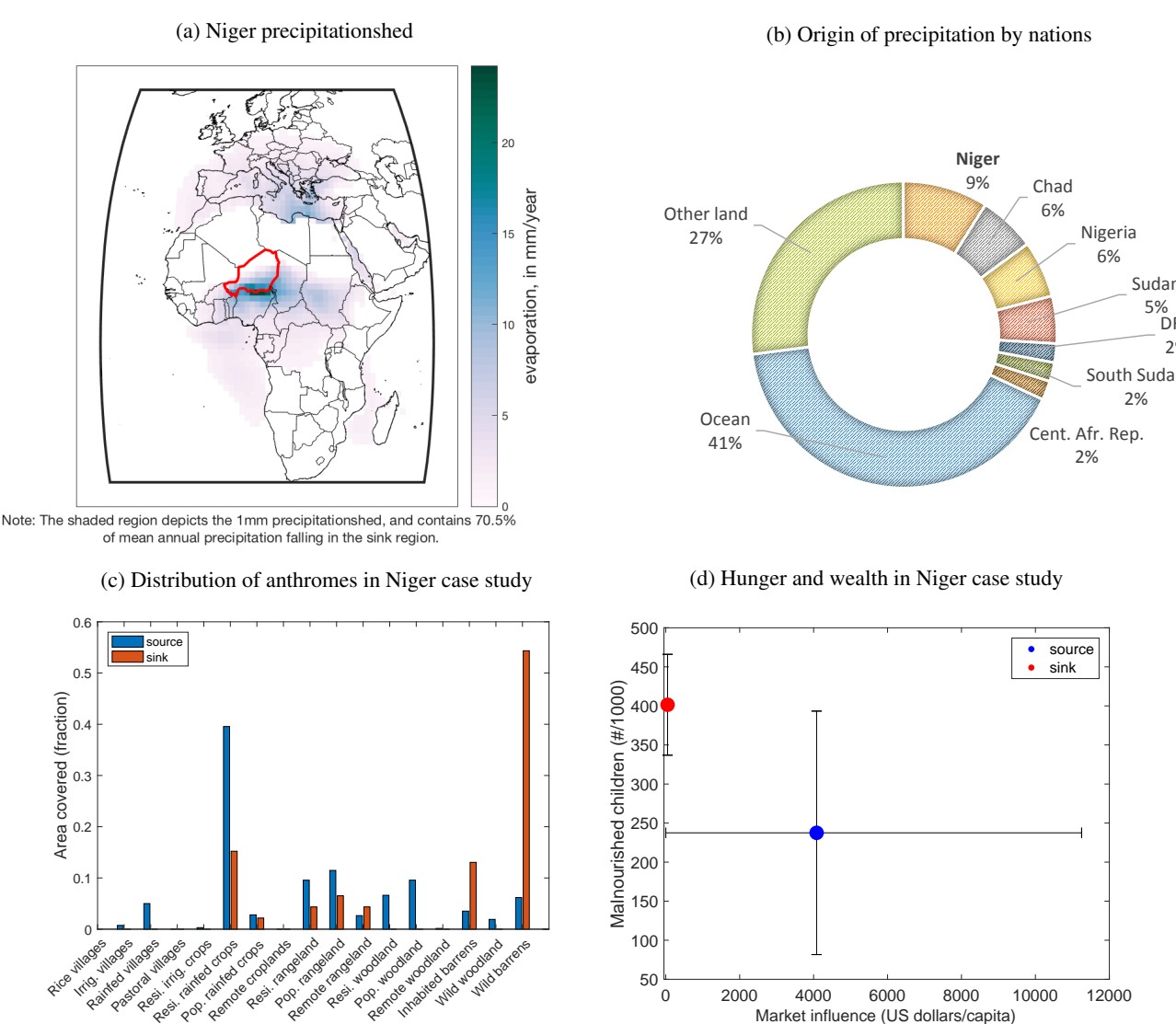

(a) Niger precipitationshed

(b) Origin of precipitation by nations

Note: The shaded region depicts the 1mm precipitationshed, and contains 70.5% of mean annual precipitation falling in the sink region.

(c) Distribution of anthromes in Niger case study

(d) Hunger and wealth in Niger case study

**Figure 3.** Niger case study including (a) precipitationshed analysis, (b) key source countries, (c) comparison of land-uses between country and precipitationsheds, and (d) comparison of economic and food security aspects of country and source regions.

This process led to large-scale confusion in part due to poorly executed policies, underfunded and understaffed government agencies, and unintended entrenchment of rural power hierarchies (Benjaminsen et al., 2009). Thus, current land-tenure in Niger is working towards clearer ownership and tenure, yet remains challenged by underfunded institutions and intractable overlapping claims of ownership. Also, Niger has not been subject to the global phenomena of land acquisitions (aka land grabs,

large-scale land acquisitions), perhaps as a result of low rainfall overall, slowly-improving land tenure, or lack of available land.

Directly to the south, Nigeria provides a considerable fraction of Niger's rainfall, and has a uniform, national land-use policy (i.e. the Land Use Act), which essentially grants the authority of land ownership to the governor of each Nigerian state (Damilola, 2017). This was originally meant to avoid problems of land speculation, overlapping or competing claims of ownership, and protection against foreign interference with land issues. However, the current issue of land-acquisitions

by foreign entities, often for large amounts of money, makes this process of land ownership vulnerable to corrupt leaders (Damilola, 2017). Currently, Nigeria has experienced many such land acquisitions, and is among the top 20 nations globally involved in land acquisitions (Osabuohien, 2014). This is relevant primarily because significant areas of land (estimated at 360,000 hectares in GRAIN (2012)), suggests extensive potential modification of the land surface, with associated impacts on moisture recycling.

Chad, located to the east of Niger, provides around 6% of moisture, but suffers from chronic poverty. Reliance on agriculture or livestock rearing provides 80% of Chadian's employment, but open access policies for land have led to over-grazing, and inadequate management has led to deforestation and desertification around dense population centers. These dynamics contribute to an uncertain and rapidly changing land-use regime in Chad (Walther, 2016; USAID, 2010). Sudan, like Chad, is also experiencing rapid land-use change, though with more-pronounced land-tenure insecurity and the ability of centralized government

to lease land without consulting local communities. Sudan has been a key target of land acquisitions, leading to internal conflict and potentially displaced persons. As with Chad, there is high potential for unpredictable land-use changes including both increased evaporation from agricultural expansion and desertification from unsustainable land and water management (USAID, 2013).

Niger's primary trading partners are France, Thailand, Malaysia, and China, with Nigeria as the only notable regional

trade partner. France is the destination and origin of greater than 30% of both exports and imports. Regionally, Nigeria is the destination of 9.5% of Niger's total export volume, primarily in the form of refined petroleum. Similarly, Niger receives about 5.8% of its total imports from Nigeria, primarily in the form of cement, electricity, and tobacco. Niger, however, represents a tiny trade partner for Nigeria, providing only 0.25% of Nigerian imports, and 0.18% of Nigerian exports. Beyond this, Chad and Sudan are very poorly integrated in terms of total trade volume with Niger and Nigeria. Thus, the countries in this region

appear to be much more economically tied to countries outside the region, than within the region. Overall, this suggests limited regional economic integration.

To conclude, the rapid land-use change taking place in many parts of Niger's precipitationshed suggest there is a high potential for change in moisture recycling driven by social-ecological processes. The ability to influence one another's land-use, and subsequently moisture recycling, is thus possible. However, active coordination among key sources in Niger's precipitation-

shed is relatively low. Some international institutions, such as the International Water Management Institute's Water Land and

Ecosystems programme, enable some trans-boundary policy coordination on key water and ecosystem issues (Saruchera and Lautze, 2015). Meanwhile, other types of activities, such as Forest Stewardship Council certifications, have considerably less influence (Nasi et al., 2012; Malhi et al., 2013). The pace of land-use change, the dense and growing dynamism of populations in all nations in Niger's precipitationshed, and the mixture of internal and external policy effectiveness suggests a medium level of social connectivity.

### 3.3 Bolivia case study

#### 3.3.1 Precipitationshed analysis

In Bolivia, the source of the precipitation is distributed across the country, with a slight concentration in the north (Fig. 4). Despite the high concentration of moisture supply sources within the country, Brazil is the most important moisture supplier thanks to both a high concentration of moisture supply just north of Bolivia's border in the Acre region, and to a low concentration of moisture supply covering Brazil's entire domain. The Peruvian Amazon is also an important contributor, as is the extreme north of Argentina.

#### 3.3.2 Social characteristics

Bolivia's anthromes are more than 50% rangelands, and a little more than 25% woodland, i.e. the Amazon. The key source areas are the Amazon in Brazil in Peru, but broadly speaking, Bolivia's precipitationshed includes a high fraction of rangelands (about 50%). Rainfed croplands comprise much of what remains (about 10%). Bolivia is characterized by relatively low child malnutrition (mean of 72%, with a standard deviation ranging from 5% to 10%), and market influence of $341, ranging from just above $0 to more than $800. This is not very wealthy compared to some regions, but this is considerably better off than other region's chid malnutrition (especially the Niger case study). Bolivia's sources of moisture are characterized by societies with higher mean malnutrition (14%) and higher wealth ($506), though the standard deviation exceeds all of Bolivia's values. This means that in terms of the standard deviation, Bolivia's social characteristics are well-within the range of the societies in its precipitationshed.

#### 3.3.3 Literature-review of social dynamics

Bolivia's precipitationshed includes key contributions from within Bolivia itself, from Brazil, and from Peru. The dominant land-uses throughout the key source regions are rangelands and forests. The strength of land-use management, in terms of governance effectiveness varies among these three nations, as does the level of land-use change, ranging from well-developed land-use methods (such as in Brazil) and much lower impact, though with high potential (as in Peru). Bolivia itself generates 18% of its own rainfall, primarily from tropical forests, the pantanal wetland, and rangelands. Historically, Bolivia's government has had a strong control on protection of forests from change, such as the first "debt-for-nature swap" in 1987 (Hansen, 1989). These, and other projects such as REDD and REDD+ projects aimed at keeping forests intact, have also been criticized

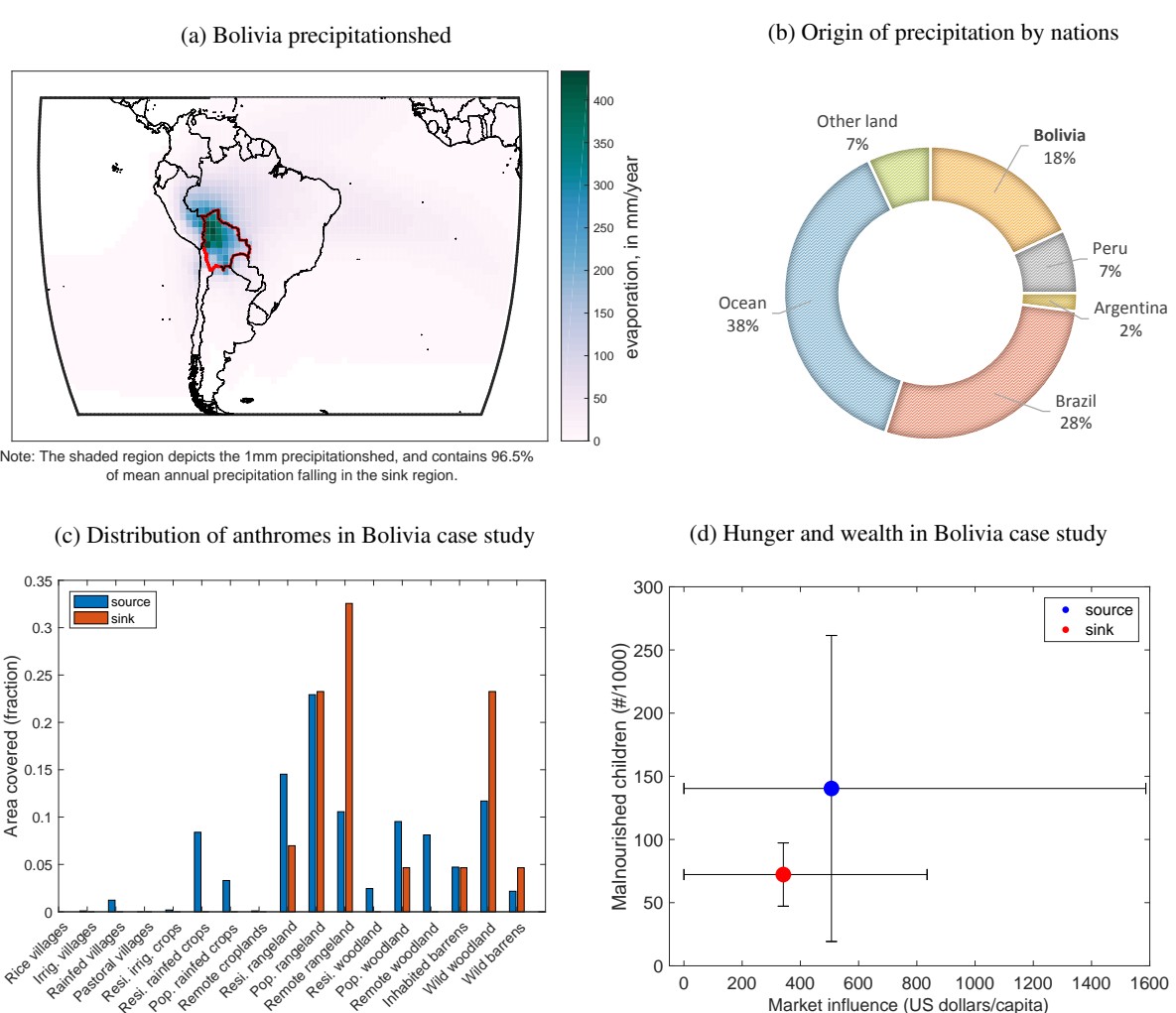

(a) Bolivia precipitationshed

Note: The shaded region depicts the 1mm precipitationshed, and contains 96.5% of mean annual precipitation falling in the sink region.

(b) Origin of precipitation by nations

(c) Distribution of anthromes in Bolivia case study

(d) Hunger and wealth in Bolivia case study

**Figure 4.** Bolivia case study including (a) precipitationshed analysis, (b) key source countries, (c) comparison of land-uses between country and precipitationsheds, and (d) comparison of economic and food security aspects of country and source regions.

for simply leading to 'leakage' of land-use change to other regions, either within Bolivia or beyond its borders (Verweij et al., 2009).

In adjacent Peru, the key forested areas that could change, and thus led to changes in moisture recycling, are very difficult to access, yet as population migration to the Peruvian Amazon is high, the current rate of deforestation is steadily increasing (Perz et al., 2005). Many overlapping jurisdictions and leases both among owners of land, as well as owners of different types of resources (e.g. timber, land, minerals, and fossil fuels), has led to contentious claims of ownership (Finer et al., 2008; Killeen et al., 2008). Additionally, recently built roads from Brazil through southern Peru will likely spur greater development of the region. A key challenge is that Brazil's land-use regulations and enforcement are stronger than Peru's, leading to leakage of deforestation activity, primarily for expansion of cattle grazing land.

Land-use policy in Brazil is quite strong, at least on paper. However, the regulatory environment is inconsistent, and the enforcement strongly depends on which political party is in power (Hurrell, 1991). According to Brazil's Forest Code, the central policy regulating deforestation, land-owners are required to keep 80% of occupied land as forest, but enforcement of this is difficult (Perz et al., 2005). Importantly, the key source regions of Bolivia's moisture are in the Acre region of Brazil which is quite remote, and under considerable existing deforestation pressure (Kainer et al., 2003). Notably, there is a stark contrast between existing deforestation patterns in Brazil (especially in Acre and Rondonia states), relative to deforestation in Bolivia and Peru. Also, deforestation in Brazil is rising again, after years of successful implementation of anti-deforestation policy.

The quality and strength of land-use policy within these three countries is strongly tied to both national-level policy, as well as participation in international land-use management efforts (e.g. REDD+), along with international trade efforts (e.g. Forest Stewardship Council certification) (Killeen et al., 2007). As a result, this region's actual land-use management exhibits a wide range of effectiveness, despite many controls in place to avoid large-scale change. Land-use change leakage (Verweij et al., 2009) is more difficult to control, thus the feedbacks of strong policies within countries may lead to other problems, especially if one of these three nations becomes more vulnerable to land-use change leakage from internal changes, due to management decisions in an adjacent country. In addition to historic trends in land-use change, recent evidence suggests a large shift is taking place in Amazonian deforestation away from Brazil to Bolivia and Peru. This is driven by several factors, including the opening up of Peru's interior via new transport networks, the moratorium on soya cultivation in Brazil, and previous deforestation estimates not accounting for small-scale and artisanal deforestation (Kalamandeen et al., 2018).

Bolivia's dominant exports are fossil fuels (45%) and minerals (zinc, precious metals, lead, gold, etc. nearly 30%) (Simoes and Hidalgo, 2011). Bolivia's dominant export to Brazil is Natural Gas (97% of exported trade flow to Brazil), and this represents 48% of Brazil's natural gas imports (Simoes and Hidalgo, 2011). In terms of trade, this is the largest trade flow between these nations, and is facilitated by pipelines connecting to Brazil. Given the dependence of Bolivia on natural gas exports to Brazil for its economy, and Brazil's dependence on Bolivian natural gas, this interdependency could be a basis for cooperation on other topics such as land-use policy around natural gas reserves and pipelines, particularly the leakage of Brazilian deforestation.

The shared issues of deforestation in Peru, Brazil, and Bolivia, as well as strong legacies of deforestation policy in Bolivia and Brazil, suggests relatively strong institutional capacity for managing change. Likewise, the economic connection, albeit in the form of natural gas pipelines connecting Bolivia to Brazil, suggests reliable economic connection (Finer et al., 2008). Additionally, the strong engagement of Bolivia's source regions in international programs targeting land-use change (e.g. REDD+, FSC) implies a social tele-coupling beyond the precipitationshed that is directly interacting with land-use policy. Finally, the international drivers of land-use change, especially in Brazil regarding soya cultivation, suggests global-scale social connections (Flach et al., 2016). Thus, we suggest Bolivia and its precipitationshed experiences strong internal social coupling, as well as global tele-coupling.

## 3.4 Construction of archetypes

Based on the results from case study analysis, we see three basic patterns of social dynamics in moisture recycling systems: (1) an isolated system archetype, dominated by internal processes; (2) a regional archetype linking adjacent countries; and, (3) a tele-coupled aechetype that links precipitation sink regions with regions outside the precipitationshed boundaries.

The core structure of each archetype is empirically grounded, given that it is well understood that land-use change directly affects evaporation, with increased vegetation typically increasing evaporation, and decreased vegetation typically decreasing evaporation (Gordon et al., 2005; Wei et al., 2013; Spracklen and Garcia-Carreras, 2015). Likewise, changes in evaporation can have direct influences on the moisture recycling that returns locally (Badger and Dirmeyer, 2015; Lawrence and Vandecar, 2015). This precipitation then provides water resources to local livelihoods, including both subsistence agriculture as well as 'off-farm ecosystem services' such as livestock forage and timber (Ojima and Chuluun, 2008; Descheemaeker et al., 2011).

How well people are doing (e.g. whether they are hungry or not) will inform the decisions they make about further modifications to the landscape (Rockström et al., 2002; Enfors, 2013), such as increasing labor and investment to maintain crop yields or foregoing labor and investment with coincident decreases in crop yield, and possibly land abandonment (Mortimore and Tiffen, 1994). Local land-use policy is formulated and implemented at least partially in response to rainfall changes. These policies are based tacitly on the confidence and knowledge on precipitation patterns and moisture recycling feedbacks, as well as on how the benefits and negative impacts are distributed among different social groups (Roncoli et al., 2002). Finally, these decisions may include further land-use change or regrowth of natural land, strengthening or weakening the feedback loop.

### 3.4.1 Isolated archetype

The 'isolated' archetype is the simplest of the proposed MRSES. In terms of social dynamics actively driving change in the precipitationshed, Mongolia is isolated. In the 'isolated' archetype, we draw attention to the fact that there are few connections or feedbacks beyond local government, nor with other regional actors (Fig 5, blue arrows). The large contributions from China and Russia are so diffuse that the social processes driving the evaporation are unable to be meaningfully discussed. Likewise, the diffuse evaporation contribution from Russia are predominantly coming from Siberian Taiga which has not experienced much land-use change. If anything there has been moderate reforestation from post-Soviet land-abandonment (Meyfroidt et al., 2016), but in the regions relevant to Mongolia this has been minimal.

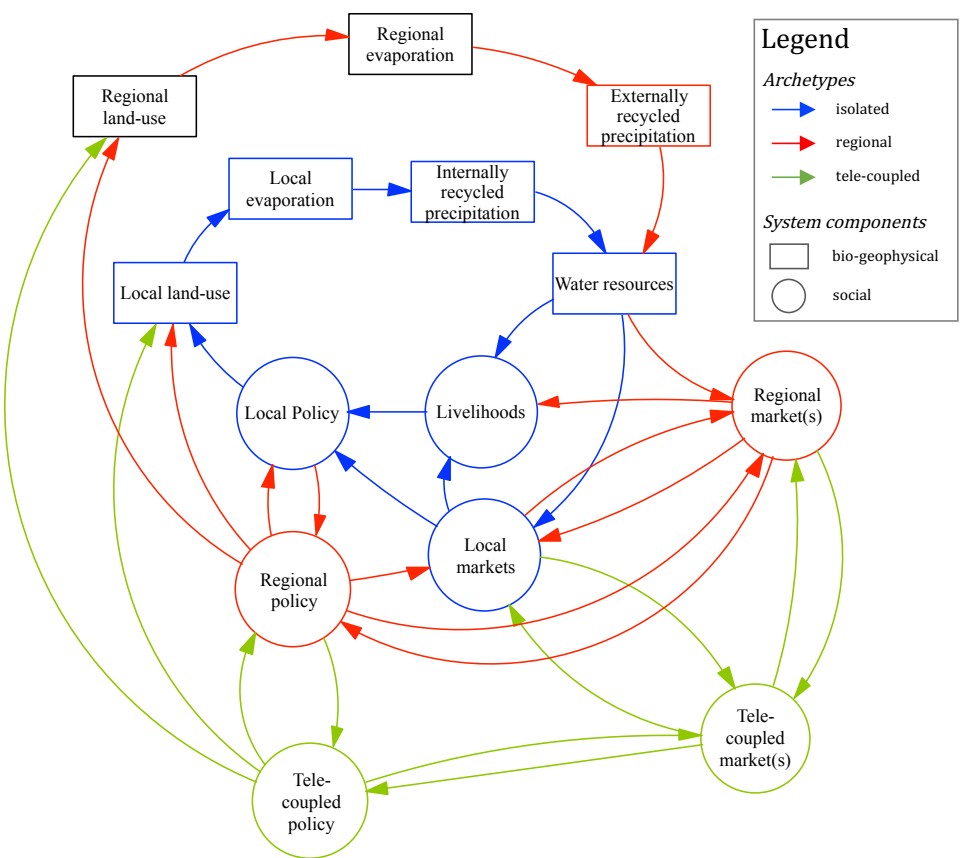

**Figure 5.** Archetypes of moisture recycling social ecological system (MRSES), with blue corresponding to 'Isolated', red to 'Regional', and green to 'Tele-coupled'.

### 3.4.2 Regional archetype

As the social connections between different sources and sinks become more numerous, regional interactions emerge (Fig 5, red arrows). Niger experiences some recycling, but also relies on contributions from neighbors in Nigeria, Chad, and Sudan. Likewise, all of these countries have active, socially-driven land-use change taking place that is impacting evaporation rates

5  (Savenije, 1995; Foley et al., 2003; Tschakert, 2007; Salih et al., 2013). The regulatory regime in these four countries varies considerably ranging from corrupt to fair and from decentralized to centralized.

To generalize, as the importance of internal moisture recycling decreases, the activities of key source regions must be considered. Where the rule-of-law is present, changes in regional evaporation will be related to government regulations or policies that serve to influence how land-use change unfolds. However, in more lawless regions where governance and institutions are

10  absent or corrupt, large-scale land-use change is typically driven by national or international corporate actors (Galaz et al., 2017). In this archetype, we see the addition of an external driver, notably 'regional policies' and 'regional markets'. Essen-

tially, we are highlighting the fact that the inner loop is now connected to regional actors, since regional land-use changes affect moisture recycling, and the subsequent social aspectxs of the core feedback loop. Notably these actors are spatially connected, e.g. within the same precipitationshed, or in adjacent countries. We also illustrate how regional interactions can more directly drive changes in external moisture recycling. Other key differences between the 'isolated' and 'regional' MRSES, are additional moisture recycling interactions with regional markets, land management, and policy. Specifically, regional and sub-regional actors have feedbacks among themselves and with 'local' livelihoods, markets, and policy nodes.

### 3.4.3 Tele-coupled archetype

The third MRSES is 'tele-coupled', and this structure draws attention to the spatially disconnected, i.e. tele-coupled, actors that can influence the social connections in the central, regional, or tele-coupled feedbacks. Bolivia, which relies on rainfall from Brazil for nearly half its rainfall would seem to be a 'regional' archetype, were it not for the dense connections to global forest conservation, deforestation driven by developed nations, and transnational agribusiness present in Bolivia, Peru, and Brazil (Galaz et al., 2015; Flach et al., 2016). These tele-coupled actors can drive land-use change in the precipitationshed, but these actors are not directly impacted by any changes in the land-use aside from perhaps changes in export commodities. Also, these tele-coupled actors are not necessarily affected by the consequences of any changes to internal moisture recycling (though they can be), while nonetheless driving changes in the MRSES itself.

## 4 Discussion

### 4.1 MRSES archetypes are idealized

Our analysis ends up with three archetypes that are idealized in structure and interaction. However, there are situations where certain aspects of the archetypes as they are represented may actually be at odds with one another. For example, in Fig 5 the policy nodes are uniformly going towards land-use. However, there is no distinguishing how Local policies might be at odds with Regional or Tele-coupled policy (e.g. local efforts to discourage deforestation could be hindered by national policies encouraging the settlement of rural forested areas). Additionally, there is greater uncertainty associated with the social nodes (Fig 5, circles) of the MRSES archetypes than with the bio or geophysical nodes (Fig 5, rectangles). This is partly because there has been much more study devoted to understanding the geophysics of moisture recycling, relative to the social dynamics of moisture recycling. Additionally, the agency of human individuals (and more broadly within societies) is a key component of the emergent complexity within MRSES.

The archetypes themselves are also idealized in that they are depicted as separate, distinct systems. Yet, in reality they are likely to exist along a spectrum from isolated to tele-coupled. Thus, for example, in moving from isolated to regional, the MRSES might first integrate the regional moisture recycling components, and then begin incorporating regional markets, regional policy, etc. The spectrum of the MRSES from isolated to tele-coupled is not meant to convey any sort of desirability one way or another; each has benefits and disadvantages. For example, in an isolated case with intense contribution from

nearby regions, there is a greater concentration of risk from accelerating feedbacks. Conversely, diffuse contribution suggests less risk from a single location, but concomitantly less ability to manage or influence land-use. In addition to this, our results may suggest that once a country crosses the threshold from being disconnected to connected to global markets it moves inexorably from being either 'isolated' or 'regional' to 'tele-coupled'. Furthermore, this dynamic is unlikely to be reversed given the momentum and increasing networked complexity of global markets and institutions, with notable exceptions, such as post-Soviet nations.

## 4.2 Advancing human-water systems understanding

The proposed MRSES framework offers a new conceptual lens to delineate system boundaries in regions where moisture recycling and human land-use decisions are substantial in comparison to other dynamics at play. This complements the various frameworks, theories, and mental models that has been developed for understanding human-water systems in terms of spatial scale, complexity of dynamics, and part of the water cycle considered (Fig 6).

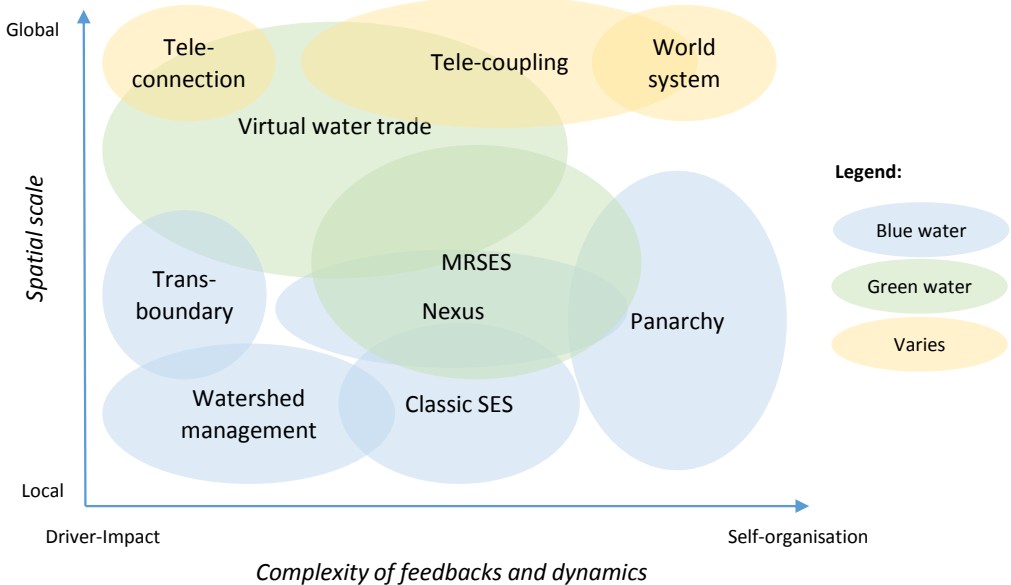

**Figure 6.** Concept of MRSES in relation to other concepts pertinent to human-water system research along the axes of spatial scale and feedback complexity considered, and in terms of water flows considered. Blue water refers to liquid water in rivers, lakes, and aquifers, whereas green water here refers to terrestrial evaporation and moisture flows.

As illustrated in Fig 6, some of the most important conceptual system boundaries considered in water management research include (increasing in scale and complexity): watershed (Giordano and Shah, 2014), transboundary river basin (Giordano and Shah, 2014), nexus approach to account for synergies across sectors (Endo et al., 2015), to concepts of water footprint and virtual water that account for the trade of embedded water (Allan, 1997). The classical SES concept (Holling, 2001; Folke et al., 2004; Folke, 2006) and panarchy (Gunderson and Holling, 2002) has typically considered social dynamics more profoundly,

but also mainly been limited to blue water (i.e., river flows, groundwater, lakes) (Gunderson et al., 2017). This is similarly true of the socio-hydrology discipline, which tends to emphasize basin-scale analyses (Sivapalan et al., 2012; Pande and Sivapalan, 2015), albeit with sophisticated, systems-thinking approaches (Elshafei et al., 2016). In land system science, teleconnection has been used to describe remote drivers of land use change, and tele-coupling has been used to describe multidirectional feedbacks (Friis et al., 2016) of which water has been considered a link through e.g., virtual water and land acquisition (Johansson et al., 2016). Other holistic concepts include world system analyses (Gotts, 2007), system integration (Liu et al., 2015), and planetary boundaries (Rockström et al., 2009; Steffen et al., 2015), which while all-encompassing, tend to have a weak local to regional perspectives of operationally policy relevant human-water interactions.

Socio-hydrological analysis similar to Fig 6 has been completed (Pande and Sivapalan, 2015) with spatial scales of socio-hydrological analysis spanning basin to regional and global scales (though global scale analyses being somewhat more limited). However, it has thus far tended to be much more on the 'driver-impact' side of the x-axis in Fig 6, rather than the 'self-organization' side of the x-axis in Fig 6, though some authors suggest there is much to be learned from complex adaptive systems (Troy et al., 2015). Hydrosocial analysis, conversely, is also focused on basin-scale perspectives, but has its roots in critical geography and Marxist theory, and is much more oriented towards articulating social hierarchies that lead to power imbalances in decision-making and equality (Wesselink et al., 2017). Thus, hydrosocial analysis is more likely to occupy the right side of the x-axis in Fig 6, and to span local to global scales, with an emphasis on the ways that power imbalances at multiple scales produce inequality in human water interactions.

Thus, the MRSES concept, while lacking the cross-sectoral or holistic perspective in comparison to nexus or world system approaches, fills a conceptual gap by accounting for social feedbacks and atmospheric moisture flows at with consideration of local to regional scale socio-economic dynamics and policy processes. Potentially, MRSES could be woven into large-scale hydrological modeling or form part of a hydro-economic model, in addition to other human interference such as irrigation, inter-basin transfer, and virtual water (Wada et al., 2017). In especially moisture recycling intense regions, MRSES could also be considered for analyzing complex co-evolutionary systems, as distilled conceptualization can be useful for exploring such dynamics (Thompson et al., 2013).

## 4.3  Systems may be reinforced in unexpected ways

The MRSES archetypes all exhibit some complexity, and also increase in complexity when moving from isolated to tele-coupled. Complexity indicates the potential for surprises induced by feedbacks (Levin et al., 2013). This is most apparent in the local and regional land-use change policies since those simultaneously have the strongest policy influence and affect regions with the highest moisture recycling values. For example, a consequence of policy decisions to graze the Kazakh steppe and then abruptly abandon this grazing, was a change in moisture recycling dynamics, and likely the rain falling in Mongolia.

A notable feature is the role of tele-coupled, spatially-disconnected actors for driving change in the precipitationshed. For example, the change in the spatial distribution of Amazonian deforestation, i.e. in Peru and Bolivia, is apparently driven by palm oil and soya cultivation in Peru and Bolivia (Kalamandeen et al., 2018). This is further emphasized in our discussion

of soya-related deforestation in Brazil, and how international agri-business interests with regard to soya cultivation, combined with Brazil's soya moratorium, are pressuring deforestation leakage into Bolivia (Flach et al., 2016; Kalamandeen et al., 2018).

Additionally, the relationships we identified as potentially existing in the system underline the reality that the system has different kinds of leverage points. For example, in the feedback loop of the isolated archetype, where policy influence and moisture recycling are tightly interconnected, there is potential for faster change but also for more immediate intervention. Conversely, the geophysically separate, socially tele-coupled drivers of land-use change can influence a region's rainfall, while the recipients of that rain have much less of an ability to influence those tele-coupled drivers of change. Moreover, the tele-coupled international actors have the potential to influence both economic policy in the sink region as well as apply market pressure to societies that are regulating rainfall. All the while these tele-coupled actors experience very little feedback from the moisture recycling system, aside from indirect changes e.g. to commodity crop price fluctuations. All of these different dynamics suggest that a portfolio of governance strategies will be necessary to address different kinds of challenges (see more on institutional challenges in Keys et al., 2017).

## 4.4 Power must be considered carefully

Though our analysis of the relationship between moisture recycling, wealth and hunger has been simplified (e.g. Fig 2d, 3d, 4d), we highlight a few key considerations with regard to power and equity in MRSES. First, in 'isolated' MRSES (e.g. Mongolia) there can still be a wide range of social characteristics (e.g. in wealth and malnutrition). The ability to influence land-use change policies that are impacting terrestrial moisture recycling may be distributed similarly. Thus, inevitably, some groups of society will have more of an ability to influence land-use change and moisture recycling, while others may simply be impacted by these changes. This sort of imbalance in control and power over resources ought to be considered.

Second, as MRSES expand to include more than one country, the potential for some parts of society within the MRSES to have more control over others may become more complex, and furthermore, the political power balance among nations become more important. In the Bolivia and Niger cases, the ability of precipitationshed nations to drive change (e.g. Brazil for Bolivia, and Nigeria for Niger) begins to matter. Moreover, in tele-coupled systems, international and non-state actors can begin driving significant terrestrial moisture recycling change, for example by interference (or control) of land-use change on commodity prices. This sort of relationship has been noted in other work as well, such as the ability for Chinese land-use decisions to impact North Korean precipitation (Wei et al., 2013).

Other work has suggested the importance of existing institutions for governing moisture recycling (Keys et al., 2017), but difficult questions have yet to be answered, e.g. "who gets to change the rain?" In many poor countries and regions, land-use change is a necessary part of subsistence and survival. Acknowledging a need for fairness, equity, and perhaps a right to modify terrestrial moisture recycling (e.g. by indigenous people) may be an equitable component of moisture recycling governance. Fundamentally, this analysis suggests that as Earth system scientists continue to make strides in understanding moisture recycling and its impacts, it will be vital to acknowledge how the benefits or costs are distributed through society. Transboundary water management, can offer useful lessons and warnings regarding tradeoffs among upstream and downstream users. For example, recent analysis of hydropower on the Mekong River has revealed major gaps in how upstream and downstream users

are considered, particularly in terms of what is equitable (Grumbine et al., 2012). The study of water justice and governance is an active discipline (Neal et al. , 2014), and if MRSES are explored further, we hope that justice and equity form an integral part of that research. Hydrosocial analysis has much to offer in this regard, given its explicit "articulation of water and social power relations", and subsequently the impact that social power imbalances can have on decision-making, and "political and material inequity" (Wesselink et al., 2017). The 'power' aspects of MRSES, and human water interactions in general, could be further improved by integration with the critical, power-oriented perspective of hydrosocial analysis.

As we move forward as a scientific community, and potentially if the concept of precipitationshed gains traction in the practitioner community, it will become increasingly important to have tools that allow us to answer questions related to justice, equity, and livelihoods. Interdisciplinary scientists, especially those trained in the natural sciences, must recognize the inherent, and potentially dangerous, pitfalls their own scientific worldview. For example, natural scientists are nearly all 'positivist', meaning they assume that every meaningful assertion ought to be scientifically verifiable and provable logically, or mathematically. This is not the worldview of much of the social science community, let alone in the practitioner community or the broader public. An awareness of the diversity of scientific worldviews is critical for successfully addressing inherently value-based questions (i.e. normative questions). At a core level introspection is prerequisite for evaluating the questions being asked and how those questions are posed.

Furthermore the natural science community needs to recognize that description of current relationships in moisture recycling (e.g. sources and sinks of moisture) are inherently charged with social import. For example, demonstrating that Brazil is very important for Bolivia's rainfall, potentially adds a matter for negotiation between the two countries, with all that entails, especially in terms of responsibility and power. Recognition of these implications is critical for natural scientists to become better interdisciplinary scholars, as well as responsible and conscientious members of society.

## 4.5 Limitations and Future Work

Evaporation can be or is actively changed through e.g., policies, cultural pressures, economic incentives, legal regimes, and treaties in the social systems, and limited by e.g., water availability, soil suitability, and energy limitation in the biophysical system. The type and nature of this 'managed evaporation' is important for understanding the entire policy and resource management space. Thus, future work could undoubtedly extend and perhaps substantiate social linkages by first identifying and quantifying managed evaporation within different administrative zones. Likewise, specific policies could be linked to these administrative zones, which could explicitly link legal, policy, and on-the-ground management efforts with particular flows of evaporation, and subsequently moisture recycling.

The analysis of moisture recycling relationships, market influence, and child malnutrition suggests a reasonable basis for exploring the social dynamics of moisture recycling more broadly. Existing work has examined the import and export of moisture among nations (Dirmeyer et al., 2009), and this work has presented an approach for developing an interdisciplinary understanding of the feedbacks within MRSES. Future work could explore the social dynamics of the specific countries with the lowest market influence and highest child malnutrition. This type of effort could align well wth research agendas targeting

the Sustainable Development Goals (SDG), and specifically with efforts to link moisture recycling to SDG achievement (Keys and Falkenmark, 2018).

# 5 Conclusions

Here, for the first time, we systematically explored the social dynamics of moisture recycling. We provide an approach, based on multiple quantitative and qualitative methods, for revealing the structure of moisture recycling social-ecological systems (MRSES). We demonstrate this approach using three case studies - Mongolia, Niger, and Bolivia - and describe the social dynamics that have the potential to impact evaporation and subsequently moisture recycling. The key conclusion is that quantitative analysis is not enough to determine which drivers are most important for the social dynamics of moisture recycling systems. A qualitative understanding, strengthened by a familiarity with relevant land-use change drivers, is critical to unraveling whether a region has social dynamics that are 'isolated', 'regional', or 'tele-coupled'. Finally, we argue that Earth system scientists need to explicitly consider the social dynamics of their work to more holistically represent reality, as well as to better engage interdisciplinary science.

*Data availability.* All data used in this paper are available from other work. The moisture recycling data for Mongolia, Niger, and Bolivia are available here: https://hdl.handle.net/10217/184640. The market influence data is associated with Verburg et al. (2011), and is available here: http://www.ivm.vu.nl/en/Organisation/departments/spatial-analysis-decision-support/Market_Influence_Data/index.aspx. The malnutrition data is from (Socioeconomic and Data Applications Center SEDAC, 2005) and is available here: http://sedac.ciesin.columbia.edu/data/set/povmap-global-subnational-prevalence-child-malnutrition. Finally, the Anthromes are from (Ellis and Ramankutty, 2008), and is available here: http://ecotope.org/anthromes/v2/data/.

*Author contributions.* Both authors contributed to conceptual development, data analysis, and manuscript preparation.

*Competing interests.* The authors disclose no competing interests.

*Acknowledgements.* We appreciate the encouragement of many academic peers to work outside our comfort zone, especially Line Gordon, Thorsten Blenckner, and Sarah Cornell.

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
