# Peer review of "On the social dynamics of moisture recycling"

_Earth System Dynamics, 2017_

## Referee Comment (RC1) · Anonymous Referee #1 · 4 Jan 2018

This study examines the social dimensions of moisture recycling taking the case of three countries: Mongolia, Niger, and Bolivia. The characteristics of sources and sinks of moisture are examined to understand the heterogeneity of moisture recycling social-ecological systems. A moisture tracking model called the Water Accounting Model-2layers (WAM-2layers) is used to track the sources and sinks of moisture starting from the moisture entering a grid cell as evaporation. The study finds that sources and sinks of moisture can experience different levels of human well-being and highlights the need to include power discontinuities in the description of moisture recycling social-ecological systems, and aims to contribute to the ongoing discussion about the emerging discipline of socio-hydrology. The paper is well written and is a good fit for Earth System Dynamics, but significant revisions should be made before the manuscript can

be considered for publication. Please find my comments below.

1. The abstract is not fully representative of the paper. First, there is no mention about what model/tool is used to carry out the analysis. And second, the abstract is too qualitative. I suggest (a) adding some information about the model; (b) adding some quantitative information about the key findings; and (c) providing some take-home message about the differences in the coupled social-hydrological systems among the three selected regions in relation to the archetypes discussed in the paper.

2. Page 2, Line 28: It is not clear of how this paper provides information for land-water managers; I didn't find any discussion in the remainder of the paper. It is important to add this information because it has been highlighted as one of the major contributions of the paper.

3. Section 2.1: The selection of the three countries for case studies is justified based on the authors' prior work and the global regions that receive significant precipitation from upwind evaporation. Given that the goal of the paper is to examine the connections between moisture recycling dynamics and social-ecological systems, wouldn't it be interesting to conduct the study in regions where there is an ongoing intensification of human activities and where hydrological-social systems are more tightly coupled and are fast evolving? In South America, the Cerrado Biome is one of such regions that is undergoing rapid land use/land cover change due to agriculture expansion. Studies have shown that the changes in land use in the Cerrado region have decreased the amount of water recycled to the atmosphere via evapotranspiration (Spera et al. 2016). There are also other regions where rainfall patterns and ET have been altered by human activities, especially land use change and irrigation (e.g., High Plains, Northwest India, Eastern China). Some of these regions also coincide with the regions of strong land-atmosphere coupling identified by Koster et al. (2004). Finally, from hydrologic point of view it would be more meaningful to conduct such a study over a river basin.

4. Page 4, Line 11: What are the necessary inputs for WAM model? What is the spatial

resolution? Please provide detailed information about the model, data, and experiment settings.

5. Page 5, Line 19: "...coarsest grid resolution": Please specify the resolution/grid size?

6. Section 3.3: This short section about the integration of moisture recycling and social features doesn't provide much information about such integration. The authors present a figure from their previous study and refer to the literature review section for further context. In the current form, I don't see this section providing any new information. I suggest the author to revise this section and make a strong case about this important integration.

7. Section 3.4: This is related to the previous comment. This rather lengthy and descriptive section provides a good literature review but it is purely qualitative and doesn't provide a good linkage with the quantitative analysis provided in other sections. A better integration of the "quantitative" and "qualitative" parts is needed.

8. Sections 3.1 and 3.5: How is land use change considered in the model? What data is used and at what resolution? Is deforestation and agricultural and irrigation expansion considered? If so, does the model account for the changes in ET because of such land use changes? Please provide these details. I also suggest that the author strengthen Section 3.1 (the quantitative analysis) by including more results from the model (e.g., results of changes in land use and the impacts on moisture recycling). Currently, this section is too brief and focuses mostly on the precipitationsheds shown in Figure 2. Please also see comment #6 on better integration.

9. Section 3.8: Please consider expanding the discussion by adding information about studying human-water interface using hydrological modeling, in line with the discussion provided by Wada et al. (2017).

10. Figure 2: This is a minor issue, but I suggest changing the unit to mm.

11. Page 8, Line 23: reference needed after "malnourished rangeland systems".

12. Page 12, Line 21: change "lead" to "led"

13. Page 12, Line 31: reference needed after "corrupt leaders".

14. Page 14, Line 31: MRSES has already been defined.

References

Koster, R. D., P. A. Dirmeyer, Z. Guo, G. Bonan, E. Chan, P. Cox, C. T. Gordon, S. Kanae, E. Kowalczyk, D. Lawrence, P. Liu, C.-H. Lu, S. Malyshev, B. McAvaney, K. Mitchell, D. Mocko, T. Oki, K. Oleson, A. Pitman, Y. C. Sud, C. M. Taylor, D. Verseghy, R. Vasic, Y. Xue, and T. Yamada, 2004: Regions of Strong Coupling Between Soil Moisture and Precipitation. Science, 305, 1138-1140.

Spera, S. A., G. L. Galford, M. T. Coe, M. N. Macedo, and J. F. Mustard, 2016: Land‐use change affects water recycling in Brazil's last agricultural frontier. Global change biology, 22, 3405-3413.

Wada, Y., M. F. Bierkens, A. De Roo, P. A. Dirmeyer, J. S. Famiglietti, N. Hanasaki, M. Konar, J. Liu, H. M. Schmied, and T. Oki, 2017: Human–water interface in hydrological modelling: current status and future directions. Hydrology and Earth System Sciences, 21, 4169.
* * *

---

## Referee Comment (RC2) · P. Dirmeyer (Referee) · 4 Jan 2018

Summary comments:

This is a noble effort to develop and demonstrate a more substantial conceptual linkage between atmospheric water cycle research (based in the "precipitationshed" concept of the lead author) and the social and economic factors in the regions linked by these hydrologic connections. However, I feel there is quite a bit of room for improvement, even in this first attempt, in terms of clarity, consistency, and organization in the presentation. While the choice of the case studies is fine to illustrate the range of archetypes defined in the MRSES structure, the presentation is uneven, as I describe below. I think it may be an issue of completeness and communication of the ideas. The authors take

on the difficult task of weaving together elements of climate, economics and social science, but are not always clear from sentence to sentence which they are talking about. It appears to be the case that the authors have become quite familiar with their own topic and forgotten how convoluted it can appear to newcomers. As a result, the paper rushes through a lot of material too quickly. More "handholding" would be appreciated! Several of the figures need improvement as well.

General comments:

A. For example, it took several readings before I really understood (I hope this is the point) that the important /input/ is how much evaporation in a precipitationshed is from managed land, inside or outside the country, and if that land is undergoing (or liable to undergo) land use change. Connectivity, the ranges described in the 3 archetypes, stems from this (right?). A plot of managed evaporation, or a table, would do wonders for clarity. One possibility would be modification to Fig 3 with a more stark color key designed along the axis of the degree of human management/impact.

B. The subsections in section 3.2 are only single paragraphs for each case. This, and the jumping around between cases later in the paper was quite jarring to this reader. I think it would be better to organize the results by presenting each case separately in its entirety, from the quantitative hydrologic and socioeconomic analysis to the social dynamics cases. Then Sec 3.5 can be the point where they are knitted back together in the framework of MRSES.

C. It would be helpful for the authors to draw the distinctions between "Market Influence" and the economic links between the case study countries and their neighbors, or even "global markets" as invoked in Sec 3.6. Without defining "Market Influence", which is really very local having only a vague implication that large cities link to global markets, there is a tendency to associate the two when they are not very related. Or do the authors, via Fig 4, try to assert that they are? This needs to be clarified.

D. An element of the social connectivity analysis eludes me. What is a more favorable

archetype to be in; isolated, regional or tele-coupled? Or are there a range of implications for each (I presume this is true)? There is a natural tendency to try to view these archetypes on a scale from bad to good. If this is not intended, the authors should proactively disabuse the readers from looking at MRSES in such a light.

E. Another detail that the authors need to spell out for unaware readers is that recycling rate (within nations) depends strongly on national area (cf. Dirmeyer and Brubaker (2007 http://dx.doi.org/10.1175/JHM557.1); the "scaled RR" in Dirmeyer et al. 2009 accounts for this). So comparing recycling among the study countries (or more generally any countries or regions) should acknowledge the strong effect of total area. Mongolia, Niger and Bolivia are in order of decreasing size and thus expected decreasing recycling rates given other factors like precipitation regime are controlled for.

Specific comments:

1. Fig 1: Please explain the red arrow at the bottom of panel C - what does this connote?

2. P5 L10-11: It is not clear to me how this notion of using spatial sampling as a proxy for temporal sampling in social data has been exploited in this study. Can you point out, perhaps retrospectively in the conclusions or /in situ/ if there is a good example, where this has been applied?

3. P5 L16: Please define Market Influence as used here. I did go to Verburg et al. (2011) to learn this, but it is a simple enough metric that it could be described here in one sentence.

4. P5 L18: Please give the specifics of the "various resolutions" of these data sets, including the time periods they each cover.

5. Table 1: Naturally I compared these values to Dirmeyer et al. (2009) [by the way, the wrong paper is listed in the References; see: http://dx.doi.org/10.1016/j.jhydrol.2008.11.016] and found the recycling and near-field

percentages in Table 1 to be generally lower. Perhaps WAM-2layers and QIBT have systematic differences in moisture advection rates?

6. P6 L6: Fig 5 is cited before Fig 4.

7. P8 L23: What is "malnourished"? As written, the rangeland systems are. I think you mean the people in those systems. Likewise in L26-27, "areas" are not hungry, the people in them are.

8. Fig 3: I think this must be mislabeled. The open circles must mark the in-country sinks and the colored dots are the various evaporation sources, no? Also, please expand or define the abbreviations "Pop." and "Resi." (actually it would be good to point out the threshold population density of 10 people/km^2 between the categories). Finally, what does the size of the circles indicate and what is the scale for that?

9. Secs 3.2.1 - 3.3.3 relative to Fig 3: It is not always evident by eye the assertions made regarding the relationships between elements in Fig 3. I think it would help to plot in each panel the first two moments (mean and standard deviation) as two crossed whiskers (along X and Y axes): one for all the source areas (weighted by contribution - is that the size of the circles?) and one for the in-country sink. Then their differences and the overlap of the ranges of standard deviations can be easily seen, and statements like P10 L3-4 and L9-10 would have a better basis.

10. Fig 4: The color bar is very unclear. Log scale? The numbers are linear, and appear to be multiplied by 10^4; clearly not what the authors intend and not commensurate with the ranges in Verburg et al. (2011).

11. Also Fig 4: In fact, I cannot see how Bolivia has such a high Market Influence index (so blue) based on the data of Verburg et al. (2011); their Fig 3 shows this to be very low for Bolivia. The colors seem to have more to do with the "qualitative" descriptions in Sec 3.4 than the quantitative data.

12. Also Fig 4: It would be good to note somewhere that the X-axis naturally corre-
lates with the size of the country (small=low) and its aridity (dry=low), while the Y-axis correlates with continentality of the climate.

13. P11 L29: "detectable changes in vegetation and associated changes in near-surface meteorology" - please provide a reference for this statement.

14. P12 L33-34: Not a sentence; appears to be missing a clause.

15. Sec 3.4.2: The Niger case is missing a discussion of the economic links among neighboring countries like exists for the other cases. This makes the final paragraph much more "hand-wavy" than the discussions of the other two cases, in my opinion.

16. Sec 3.4.3: Much is made about the strength of national land use regulations, but (1) in the particular case of Brazil they are highly variable in time, depending on which party is in power; (2) enforcement lags behind (this is discussed somewhat) and (3) the spatial and population scale of the problem makes such statements about regulation almost meaningless. The Acre region of Brazil, noted as a main external moisture source for the Bolivian precipitationshed, has experienced significant deforestation over the last 40 years, albeit not as widespread as Rondonia, which also borders Bolivia. The problem seems to be soft-peddled a bit here.

17. P14 L11-12: Likewise, the notion that the "region's land use is relatively well-governed with many controls in place to avoid large-scale change" seems untrue, and in contrast to the very next sentence. It is clear even from Google Maps that there is a very clear demarcation following the Bolivian border where deforestation is rampant in neighboring Brazil right up to the border.

18. Sec 3.5: In addition to explaining the archetypes and how they fit the previously presented data and social dynamics review, it should be frankly stated where they are unclear, or at odds.

19. Fig 5: I also find this diagram somewhat unsatisfying, but perhaps I am not understanding it. Are the boxes meant to be static, or is it the fluctuations (changes)

in the boxes that precipitate (pardon the pun) effects in other boxes by the arrows? For instance, Mongolia has a huge non-local (regional) evaporation source, small (9%) local source, and strong export connections to its large neighbors, yet is defined as "isolated" based on other factors. The synthesis and the weights given to the various factors seems either subject to interpretation, or not clearly enough defined. If the single driving /hydrologic/ factor is land use change in the precipitationshed, it could be demonstrated much more clearly and succinctly than has been done here (see General Comment above). Then it comes to the economic consequences to fill out the MRSES archetypes - am I seeing this correctly?

20. P15 L6-7: "...large contributions..." of what? Be clear and complete. "...social processes driving the evaporation..." - this statement may be endemic of the communication problem. Presumably this is shorthand for "policy and economics drive land use change that affects evaporation potentially affecting downstream precipitation" but I think the authors forget how much better they understand their own material than their readers will.

21. P16 L5-6: This needs to be stated earlier, to clarify much of what goes before.

22. P16 L33-34: It is sentences like this that lead to confusion; the "regional factors that can influence...." means land use changes affecting moisture sources, right? If so, just say that. I think the hydrologic underpinnings get lost at times in this manuscript.

23. P17 L8: I think this is a typo: "and Brazil" should be "from Brazil"

24. Sec 3.5.3: Aren't the actual drivers of land use change (deforestation in the Amazon) much more from developed nations than is the case in Niger? Doesn't this also have implications for "tele-coupling", or not as defined?

25. P17 L28-30: This sentence would benefit from a concrete example or pointer to the specific data presented earlier.

26. P18 L1-2: Similarly, this sentence would benefit from an actual example of reinforcement/surprise, and not merely describing the situation.

27. Sec 3.8: This section I found to be more clearly presented. I am reminded of the interesting case evident in Wei et al. (2013; http://dx.doi.org/10.1175/JHM-D-12-079.1) where evaporation from irrigation in Northeast China appears to supply a significant amount of rainfall to North Korea!

28. P19 L9: "...gives Brazil power over Bolivia in potentially significant ways." I would phrase it that it gives Brazil "responsibility to Bolivia" - this is the other side of the coin from air pollution (including nuclear fallout), where it is pretty easy to track sources to those affected downstream. We are not used to thinking of water vapor in that way, but "responsibility" gives a more overarching concept to such linkages.

—————————————————————

---

## Referee Comment (RC3) · Anonymous Referee #3 · 5 Jan 2018

Overview: The paper by Keys et al presents three case studies of links between the social and terrestrial moisture recycling system. This study combines quantitative modeling of terrestrial moisture recycling with metrics and a literature review of social factors. In this way, the study estimates the major sources of precipitation (i.e. precipitation-sheds) for three case study countries. Gridded social variables are then evaluated for the source and sink nodes in each case study. Finally, a literature review is performed to reveal additional context for each case study and enable the development of moisture recycling social ecological systems archetypes. Overall, I think this is an innovative, well-executed, and (reasonably) well-written paper that would make a unique contribution to the literature. I recommend publication after consideration of my comments below.

[Figure]

Major comments:

1. Human well-being/welfare has a precise definition in the social sciences literature.

The term(s) "well-being" and "welfare" are used several times in the paper. I don't think these is the best term to use, since they mean something precise in the economics literature that is distinct to the meaning here. I think it would be better to refer to "social" aspects/variables/indicators of source and sink nodes. Then, the precise metric references should be specified whenever possible.

2. Why not perform a global scale analysis?

The literature review would be too difficult to perform for all countries in the world. However, a global scale analysis of precipitation-sheds and receiving countries would be relatively straightforward to perform. It appears the authors have all the information they need for this. They have WAM-2 pixels, social variables at the pixel scale. So, couldn't this be a global scale analysis for most aspects? Then, the 3 case study countries could be used for the literature review portion of the paper.

If a global scale analysis is performed, then the authors will have more data to run some interesting regressions. For example, they can calculate "precipitation-sheds" and "sink" nodes for all countries. Then, they can obtain average values of social variables in each source/sink. In this way, they will have enough statistical power to run multivariate regressions of the driving factors of the terrestrial moisture recycling system.

3. Fig 3 is confusing and could be simplified.

There is a lot of information in Fig 3. I don't think most of it is necessary. For example, does the biome information convey anything interesting? There does not appear to be any trend between malnourished children (y-axis) and GDP/capita (x-axis), so this information could be made easier to read. I think this figure would be better if it presented the average value of malnourishment and GDP/capita explicitly for the source and sink

region of each country. This might be able to be accomplished with a simple bar graph or box-whisker plot for each variable for each source/sink node. A table might even best illustrate upstream/downstream differences. This simplicity would better illustrate the main points made in sections 3.2.1-3.2.3.

4. The section on power dynamics could be improved.

There seem to be many similarities between upstream/downstream power dynamics in precipitation-sheds and watersheds. I think this section would benefit from drawing from the power dynamics concepts in the transboundary watershed literature. A lot of work has been done on power/politics in international river basins that section 3.8 would benefit from referencing.

Generally, section 3.8 could use a bit of a rewrite for clarity. Have any papers quantified the impact of upstream precipitation-sheds on downstream droughts? This seems like it would be the most clear example of upstream-downstream conflict/power issues. Also, can you expand on the Daw et al (2011) reference? Does this paper specifically focus on power dynamics in teleconnected systems?

5. A bit more connection with the SES and socio-hydrology literature would be helpful.

How does this work relate to socio-ecological systems (SES) work? Have similar archetypes (Fig 5) been presented in SES literature? Or socio-hydrology? What outcome variables are primarily of interest in the SES literature?

Minor comments:

1. The term "average market influence" is not clear and confusing. Please just call it what it is, i.e. GDP per capita. 2. Figure 4 doesn't seem to show much. What happens if you just plot national international moisture recycling (y-axis) against GDP per capita (x-axis)? 3. P 18 line 14: "Though the analysis of environmental justice flows has been simplified (Fig 3)….". Environmental justice flows are not quantified or presented in Fig 3. This statement is not warranted. 4. P 19 line 5: This sentence is a bit ironic. It

seems to be a call for interdisciplinary scientists to engage and communicate with one another. However, this sentence is laced with jargon that is not widely understood (i.e. "positivism", "normative terminology").

---

## Referee Comment (RC4) · Anonymous Referee #4 · 16 Jan 2018

Major comments

The authors investigated terrestrial moisture recycling in three inland countries, namely, Mongolia, Niger, and Bolivia, by focusing on land-use change in moisture-source regions. By investigating land-use change policy of the countries in question and surrounding countries, the authors tried to explore the social dynamics of moisture recycling. Although I found the attempt quite interesting and novel, the manuscript in present form lacks clarity and quantitative evaluations in many parts. Hope the comments below are useful for further improvements.

Specific comments

Page 1 line 8 "We find that the sources and sinks of moisture can experience very different levels of human well-being, suggesting that power discontinuities must be included in the description of MRSES dynamics": How moisture "can experience different levels of human well-being"? What are "power discontinuities"?

Page 1 line 11 "This exploration of the social dimensions of moisture recycling": It seems an important precondition of this work that the "social dimension" plays an important role in terrestrial moisture recycling, but this is hardly proved (quantified) in text. I suppose the direct impacts of land-use change on the terrestrial hydrological cycle would be marginal. Exceptions are the cases for quite intensive irrigation (e.g. DeAngelis et al. 2010; Puma and Cook, 2010) and land-use change at continental- and century-scale (e.g. Takata et al. 2009).

Page 4 line 3 "2.2 Tracking the sources of moisture": The authors applied the WAM-2layers model to estimate the evaporation and precipitation of their study domain. First, I would suggest providing more detailed information on the boundary condition (i.e. simulation period, land-use assumption, validation data). Second, I would suggest conducting some additional simulations under counterfactual land-use which implies historical land-use change mentioned in Section 3.4. Such simulations would be highly effective to convince readers how significantly "social dynamics" would change precipitation or evaporation.

Page 8 line 26 "in general evaporation arising from relatively wealthier, less hungry areas falling out as precipitation in poorer hungrier areas": This part sounds very subjective. Add figures and tables to make this part quantitative and concrete.

Page 10 line 3 "However there is a flow of moisture from wealthier areas to poor areas (relative)": Same comment as above.

Page 10 line 7 "Within Bolivia itself, there is a cluster of wealthier rangelands and populated woodlands, and a cluster of much poorer remote and wild forest systems" Same comment as above. What is a cluster?

Page 10 line 8 "Surprisingly": Explain what is surprising. The authors tend to connect factor and factor subjectively. What are the solid knowledge based on established evidence here? In what sense surprising?

Page 11 line 10 "affect moisture recycling policy": What is moisture recycling policy? In my view, the impact on moisture recycling is one of many (often unintended) secondary-impacts of land-use/industrial policy.

Page 14 line 29 "Construction of archetypes": Although it is an interesting idea that inland moisture recycling could be subdivided into three categories, I'm wondering how to find thresholds among them. Any region is neither fully isolated nor fully teleconnected. What to do with regions in between?

Page 18 line 15 "in isolated systems (e.g. Mongolia) there can still be a wide range of well-being (e.g. wide range in poverty and malnutrition)": I couldn't follow the authors' logic. In every isolated systems the authors' claim holds true? Which figures/tables/sub-sections clearly do clearly support this claim?

Figure 3: Very hard to understand. What does each plot represent (grid cells of each nation or those for each precipitationshed)? Also clearly indicate in text what we should focus on. These panels look random scatter without meaningful information at first glance.

References

DeAngelis, A., Dominguez, F., Fan, Y., Robock, A., Kustu, M. D., and Robinson, D.: Evidence of enhanced precipitation due to irrigation over the Great Plains of the United States, J. Geophys. Res., 115, D15115, 2010.

Puma, M. J., and Cook, B. I.: Effects of irrigation on global climate during the 20th century, J. Geophys. Res., 115, D16120, 2010.

Takata, K., Saito, K., and Yasunari, T.: Changes in the Asian monsoon climate during 1700-1850 induced by preindustrial cultivation, P. Natl. Acad. Sci. USA, 106, 9586-

9589, 10.1073/pnas.0807346106, 2009.

---

## Editor Comment (EC1) · M. Sivapalan (Editor) · 3 Feb 2018

The paper has received several critical but constructive comments. I strongly encourage the authors to respond to each reviewer, including an articulation of how they intend to address relevant comments and criticisms in a revised manuscript

---

## Author Comment (AC1) · 7 Mar 2018

Reviewer Comment = RC Author Comment = AC

RC: This study examines the social dimensions of moisture recycling taking the case of three countries: Mongolia, Niger, and Bolivia. The characteristics of sources and sinks of moisture are examined to understand the heterogeneity of moisture recycling socialecological systems. A moisture tracking model called the Water Accounting Model2layers (WAM-2layers) is used to track the sources and sinks of moisture starting from the moisture entering a grid cell as evaporation. The study finds that sources and sinks of moisture can experience different levels of human well-being and highlights the need to include power discontinuities in the description of moisture

recycling socialecological systems, and aims to contribute to the ongoing discussion about the emerging discipline of socio-hydrology. The paper is well written and is a good fit for Earth System Dynamics, but significant revisions should be made before the manuscript can be considered for publication. Please find my comments below.

AC. We appreciate the careful consideration the Reviewer has given to the manuscript. We hope that our revised manuscript merits further consideration for publication in ESD.
- - -

RC1.  The abstract is not fully representative of the paper.  First, there is no mention about what model/tool is used to carry out the analysis.  And second, the abstract is too qualitative. I suggest (a) adding some information about the model; (b) adding some quantitative information about the key findings; and (c) providing some take-home message about the differences in the coupled social-hydrological systems among the three selected regions in relation to the archetypes discussed in the paper.

AC1:  Thank you for these comments.  We have made the suggested changes, adding brief information about the analysis, some quantitative results, and more clearly articulated key messages contrasting the different MRSES.

RC2.  Page 2, Line 28: It is not clear of how this paper provides information for land-water managers; I didn't find any discussion in the remainder of the paper. It is important to add this information because it has been highlighted as one of the major contributions of the paper.

AC2.  Thank you for the comment.  We recognize that we were unclear where this text appears. We now explicitly direct the reader to the sections "4.2 Guidelines

for constructing MRSES", "4.3 Advancing human-water systems understanding", "4.4 Systems may be reinforced in unexpected ways", and "4.5 Power must be considered carefully".These four sections contain the information we originally referred to that could help land-water managers.

RC3. Section 2.1: The selection of the three countries for case studies is justified based on the authors' prior work and the global regions that receive significant precipitation from upwind evaporation. Given that the goal of the paper is to examine the connections between moisture recycling dynamics and social-ecological systems, wouldn't it be interesting to conduct the study in regions where there is an ongoing intensification of human activities and where hydrological-social systems are more tightly coupled and are fast evolving? In South America, the Cerrado Biome is one of such regions that is undergoing rapid land use/land cover change due to agriculture expansion. Studies have shown that the changes in land use in the Cerrado region have decreased the amount of water recycled to the atmosphere via evapotranspiration (Spera et al. 2016). There are also other regions where rainfall patterns and ET have been altered by human activities, especially land use change and irrigation (e.g., High Plains, Northwest India, Eastern China). Some of these regions also coincide with the regions of strong land-atmosphere coupling identified by Koster et al. (2004). Finally, from hydrologic point of view it would be more meaningful to conduct such a study over a river basin.

AC3. Thank you for these reflections. We were careful about the regions we selected, in that we wanted to ensure that we picked regions that were experiencing significant vegetation-regulated moisture recycling, which we used as a proxy for potential impacts from land-use change. Likewise, the countries are all quite similar in national area, providing a reasonable control on the spatial scale of the corresponding precipitationsheds. Also, for our purposes of emphasizing social

dynamics, countries are more suitable than hydrologic basins. This is because: (a) Many regions experience limited runoff, suggesting the most meaningful hydrological flows are not river basins but sources and sinks of atmospheric moisture (for more on this see Weiskel et al. 2014); and (b) Countries provide a lens for evaluating social dynamics, especially land-use policies, since governance institutions (e.g. regulatory frameworks, transboundary legal arrangements, etc) are typically based on political or administrative units, such as countries, rather than hydrologic basins.

RC4. Page 4, Line 11: What are the necessary inputs for WAM model? What is the spatial resolution? Please provide detailed information about the model, data, and experiment settings.

AC4. We agree that it is important for the reader to have this information available. The inputs are described in Section 2.3 "As input to the WAM-2layers, we use ERA-Interim Reanalysis data, from the European Center for Mesoscale Weather Forecasting (Dee et al., 2011). We downloaded global, model-level data at the 1.5 deg. by 1.5 deg. Resolution. The WAM-2layers uses 6-hourly data for horizontal and vertical wind, humidity, and surface pressure; and it uses 3-hourly data for evaporation and precipitation."

There are no experiments, as such, since we use the WAM-2layers to calculate how moisture moves around the planet. However, we do explain how the WAM-2layers calculates this movement of moisture in Section 2.2. We hope that this explanation eliminates any confusion regarding the functioning of the WAM-2layers.

Additionally, one of the other Reviewers commented on the need for more detail on the land-use change simulations/experiments. This is clearly a failing in our (the authors)

communication, since there were no such experiments. In this regard, we have added clarification text in Section 2.2 of the revised manuscript to clear up this confusion:

"We emphasize that the WAM-2layers is a moisture tracking scheme, and not a simulation. It is possible to couple the WAM-2layers with dynamic simulations of land-surface hydrology, including vegetation (e.g. Wang-Erlandsson et al., 2014; Keys et al., 2016), but that is not what we have done in this research. Thus, the results that we present are purely based on the implicit hydrological information contained within the ERA-Interim Reanalysis data."

RC5. Page 5, Line 19: ". . .coarsest grid resolution": Please specify the resolution/grid size?

AC5. Thank you for the comment, you and the other Reviewers requested this. We have included a table that summarizes the different datasets, including variable name, description of variable, source resolution, units, time period of analysis, and source reference. This table is found in the Methods section.

RC6. Section 3.3: This short section about the integration of moisture recycling and social features doesn't provide much information about such integration. The authors present a figure from their previous study and refer to the literature review section for further context. In the current form, I don't see this section providing any new information. I suggest the author to revise this section and make a strong case about this important integration.

AC6. Thank you for the comment, and we agree that this section needs to be revised. We have changed the format of the results presentation, based on Reviewer

2's suggestion of reporting each individual case study in its entirety. Thus, when we now present each case study, they are more coherent, with the results for the precipitationshed, social characteristics, and literature review of social dynamics presented in sequence.

Additionally" we have removed Fig 4 since it did not substantially improve the paper, and apparently led to confusion among the other Reviewers.

RC7. Section 3.4: This is related to the previous comment. This rather lengthy and descriptive section provides a good literature review but it is purely qualitative and doesn't provide a good linkage with the quantitative analysis provided in other sections. A better integration of the "quantitative" and "qualitative" parts is needed.

AC7. Thank you for this suggestion. We agree that better integration among the different threads of each cast study is necessary. With the new format of each case study presented in its entirety, we hope this addresses this issue.

RC8. Sections 3.1 and 3.5: How is land use change considered in the model? What data is used and at what resolution? Is deforestation and agricultural and irrigation expansion considered? If so, does the model account for the changes in ET because of such land use changes? Please provide these details. I also suggest that the author strengthen Section 3.1 (the quantitative analysis) by including more results from the model (e.g., results of changes in land use and the impacts on moisture recycling). Currently, this section is too brief and focuses mostly on the precipitationsheds shown in Figure 2. Please also see comment 6 on better integration.

AC8. We apologize for the confusion on this issue, since it was clearly a failure in communication. As we stated in response to comment 4, there were no land-use

change simulations in this analysis. We ran a single analysis of moisture recycling, using ERA-Interim data, which is output from the European Center for Mesoscale Weather Forecasting (ECMWF) forecast model. Thus, our analysis examines the observed record, with implicitly historical land-cover (i.e. inferred from various observational datasets).

RC9. Section 3.8: Please consider expanding the discussion by adding information about studying human-water interface using hydrological modeling, in line with the discussion provided by Wada et al. (2017).

AC9.Thank you for this suggestion, and this is discussed in detail in section 4.3 "Advancing human-water systems understanding"

RC10. Figure 2: This is a minor issue, but I suggest changing the unit to mm.

AC10. Thank you, we have made this change.

RC11. Page 8, Line 23: reference needed after "malnourished rangeland systems".

AC11. There is no reference for this, because it is based on our own analysis within this paper. Results from our case study analysis, including the analysis of land-uses and social characteristics of the sources and sinks of moisture, are more comprehensively reported in the results section. Likewise, we have re-written much of the literature review text so our statements ought to be much clearer to the reader.

RC12. Page 12, Line 21: change "lead" to "led"

AC12. Thanks for this suggestion. We have identified all past tense forms of "lead" and changed them to "led".

RC13. Page 12, Line 31: reference needed after "corrupt leaders".

AC13. Thanks, the corresponding reference has been added.

RC14. Page 14, Line 31: MRSES has already been defined.

AC14. We recognize that we already defined this, but given that it was defined much earlier in the manuscript, we decided to remind the reader here (at the beginning of the actual MRSES discussion) to ensure the reader does not now need to hunt through the paper for the definition.

References
Dee, D. P., Uppala, S. M., Simmons, A. J., Berrisford, P., Poli, P., Kobayashi, S., ... Bechtold, P. (2011). The ERA‐Interim reanalysis: Configuration and performance of the data assimilation system. Quarterly Journal of the royal meteorological society, 137(656), 553-597.

Keys, P. W., Wang-Erlandsson, L., Gordon, L. J. (2016). Revealing invisible water: moisture recycling as an ecosystem service. PloS one, 11(3), e0151993.

Koster, R. D., P. A. Dirmeyer, Z. Guo, G. Bonan, E. Chan, P. Cox, C. T. Gordon, S. Kanae, E. Kowalczyk, D. Lawrence, P. Liu, C.-H. Lu, S. Malyshev, B. McAvaney, K. Mitchell, D. Mocko, T. Oki, K. Oleson, A. Pitman, Y. C. Sud, C. M. Taylor, D. Verseghy, R. Vasic, Y. Xue, and T. Yamada, 2004: Regions of Strong Coupling Between Soil Moisture and Precipitation. Science, 305, 1138-1140.

Spera, S. A., G. L. Galford, M. T. Coe, M. N. Macedo, and J. F. Mustard, 2016: Land-use change affects water recycling in Brazil's last agricultural frontier. Global change biology, 22, 3405-3413.

Wada, Y., M. F. Bierkens, A. De Roo, P. A. Dirmeyer, J. S. Famiglietti, N. Hanasaki, M. Konar, J. Liu, H. M. Schmied, and T. Oki, 2017: Human–water interface in hydrological modelling: current status and future directions. Hydrology and Earth System Sciences, 21, 4169.

Wang-Erlandsson, L., Bastiaanssen, W. G., Senay, G. B., van Dijk, A. I., Guerschman, J. P., Keys, P. W., Gordon, L. J., Savenije, H. H. (2016). Global root zone storage capacity from satellite-based evaporation. Hydrology and Earth System Sciences, 20(4), 1459.

Weiskel, P. K., Wolock, D. M., Zarriello, P. J., Vogel, R. M., Levin, S. B., Lent, R. M. (2014). Hydroclimatic regimes: a distributed water-balance framework for hydrologic assessment and classification. Hydrology and Earth System Sciences Discussions, 11, 2933-2965.

---

## Author Comment (AC2) · 7 Mar 2018

Reviewer Comment = RC   Author Comment = AC

RC Summary comments:
This is a noble effort to develop and demonstrate a more substantial conceptual linkage between atmospheric water cycle research (based in the "precipitationshed" concept of the lead author) and the social and economic factors in the regions linked by these hydrologic connections. However, I feel there is quite a bit of room for improvement, even in this first attempt, in terms of clarity, consistency, and organization in the presentation. While the choice of the case studies is fine to illustrate the range of archetypes defined in the MRSES structure, the presentation is uneven, as I describe

below. I think it may be an issue of completeness and communication of the ideas. The authors take on the difficult task of weaving together elements of climate, economics and social science, but are not always clear from sentence to sentence which they are talking about. It appears to be the case that the authors have become quite familiar with their own topic and forgotten how convoluted it can appear to newcomers. As a result, the paper rushes through a lot of material too quickly. More "handholding" would be appreciated! Several of the figures need improvement as well.

AC. We are pleased that the Reviewer has taken a considerable amount of time to both interpret our paper, and to provide detailed feedback on how to improve the work. We hope that our responses below will both answer the lingering questions and address the problems identified by the Reviewer.
- - -

General comments:
RC-A. For example, it took several readings before I really understood (I hope this is the point) that the important /input/ is how much evaporation in a precipitationshed is from managed land, inside or outside the country, and if that land is undergoing (or liable to undergo) land use change. Connectivity, the ranges described in the 3 archetypes, stems from this (right?). A plot of managed evaporation, or a table, would do wonders for clarity. One possibility would be modification to Fig 3 with a more stark color key designed along the axis of the degree of human management/impact.

AC-A. Thank you for the comment, and we appreciate that the Reviewer spent so much time attempting to glean this insight. Ultimately, 'managed evaporation' (as the reviewer puts it) or evaporation that can be or is actively changed is the implicit focal point of the biophysical system. If the Reviewer implies that 'managed' includes all the policies, cultural pressures, economic incentives, legal regimes, treaties, etc. then yes, 'managed evaporation' is the key input. While we think there could be merit

in a table of managed evaporation, estimating this for all the systems, is an academic task in its own right.

Our goal in this paper is to explore potential ways for characterizing the social connections that link the recipient of precipitation back to the sources of evaporation: i.e., (a) describing conceptual and actual social linkages among sinks of precipitation and sources of evaporation, and (b) developing a method for linking existing methods of quantitative moisture recycling analysis (e.g. precipitationshed calculation), with new methods of quantification (e.g. relating well-being indicators to moisture recycling sinks and sources), and qualitative analysis of economic and social policies related to land-use change.

We recognize the value of a table that explores managed evaporation, so we include the following text in the new section "4.6 Limitations":
"Evaporation can be or is actively changed through e.g., policies, cultural pressures, economic incentives, legal regimes, and treaties in the social systems, and limited by e.g., water availability, edaphic suitability, and energy limitation in the biophysical system. The type and nature of this manageable or managed evaporation is important for understanding the management space. Thus, future work could undoubtedly extend and perhaps substantiate social linkages by first identifying and quantifying managed evaporation within different administrative zones. Likewise, specific policies could be linked to these administrative zones, which could explicitly link legal, policy, and on-the-ground management efforts with particular flows of evaporation, and subsequently moisture recycling."

RC-B. The subsections in section 3.2 are only single paragraphs for each case. This, and the jumping around between cases later in the paper was quite jarring to

this reader. I think it would be better to organize the results by presenting each case separately in its entirety, from the quantitative hydrologic and socioeconomic analysis to the social dynamics cases. Then Sec 3.5 can be the point where they are knitted back together in the framework of MRSES.

AC-B. Thank you for the comment, and we have used your suggested changes. We now present each case in its entirety, including (a) precipitationshed analysis, (b) land use distributions among the sources and sink, (c) social linkages between sources and sinks, and (d) literature review of policy and management.

RC-C. It would be helpful for the authors to draw the distinctions between "Market Influence" and the economic links between the case study countries and their neighbors, or even "global markets" as invoked in Sec 3.6. Without defining "Market Influence", which is really very local having only a vague implication that large cities link to global markets, there is a tendency to associate the two when they are not very related. Or do the authors, via Fig 4, try to assert that they are? This needs to be clarified.

AC-C. We agree that Market Influence needs to be better defined. Market Influence is calculated by multiplying normalized travel time to cities and ports, with national level per capita GDP in terms of purchasing power parity. Given that Verburg et al. (2011) goes into the nuanced relationship between market influence, wealth, and connection to global markets, it seems a bit redundant to do that again. We do cite Verburg to ensure it is clear where to explore those additional arguments.

RC-D. An element of the social connectivity analysis eludes me. What is a more favorable archetype to be in; isolated, regional or tele-coupled? Or are there a range

of implications for each (I presume this is true)? There is a natural tendency to try to view these archetypes on a scale from bad to good. If this is not intended, the authors should proactively disabuse the readers from looking at MRSES in such a light.

AC-D. Thank you and we agree with the Reviewer. It would be wrong to conclude that one archetype is preferred to another and that there are (as the Reviewer suggests) a range of implications for each. We explore this and other aspects of the MRSES in section 4.1 "MRSES archetypes are idealized".

RC-E. Another detail that the authors need to spell out for unaware readers is that recycling rate (within nations) depends strongly on national area (cf. Dirmeyer and Brubaker (2007 http://dx.doi.org/10.1175/JHM557.1); the "scaled RR" in Dirmeyer et al. 2009 accounts for this). So comparing recycling among the study countries (or more generally any countries or regions) should acknowledge the strong effect of total area. Mongolia, Niger and Bolivia are in order of decreasing size and thus expected decreasing recycling rates given other factors like precipitation regime are controlled for.

AC-E. We appreciate this detail, and for mentioning "scaled RR". We do not scale the recycling ratios in this paper, as the biophysical reason for low or high recycling ratios have limited relevance for the social implications for individual nations. In fact, smaller nations already tend to be more reliant (or be subject to) outside influence beyond atmospheric moisture connections: e.g., in terms of global markets, national security, and climate change impacts.

Specific comments:
RC1. Fig 1: Please explain the red arrow at the bottom of panel C - what does this

connote?

AC1. Thanks for the comment, this arrow is simply highlighting the social connection that is often missing from depictions of sources and sinks in moisture recycling research. The arrow is red to match the red boxes containing the "Social" node, and points in both directions indicating that social links can connect in multiple directions. We have clarified this in the updated caption to Fig 1.

RC2. P5 L10-11: It is not clear to me how this notion of using spatial sampling as a proxy for temporal sampling in social data has been exploited in this study. Can you point out, perhaps retrospectively in the conclusions or /in situ/ if there is a good example, where this has been applied?

AC2. We agree that this is unclear. In the revised version of the paper the sentence is no longer relevant, so it has been removed.

RC3. P5 L16: Please define Market Influence as used here. I did go to Verburg et al. (2011) to learn this, but it is a simple enough metric that it could be described here in one sentence.

AC3. We agree with the Reviewer, and as indicated in previous comments, we have expanded the definition of all variables in section 2.4 and Table 2.

RC4. P5 L18: Please give the specifics of the "various resolutions" of these data sets, including the time periods they each cover.

AC4. Thanks for the comment, and this feedback was echoed by other Reviewers. We have included this information in Table 2, which summarizes the different datasets, including variable name, description of variable, source resolution, units, time period of analysis, and source reference. This table is found in the Methods section.

RC5. Table 1: Naturally I compared these values to Dirmeyer et al. (2009) [by the way, the wrong paper is listed in the References; see: http://dx.doi.org/10.1016/j.jhydrol.2008.11.016] and found the recycling and near-field percentages in Table 1 to be generally lower. Perhaps WAM-2layers and QIBT have systematic differences in moisture advection rates?

AC5. Thank you for comparing the values, and for noticing this error in the citation. Many apologies! It could be that we misinterpret the comment, but we ought to be comparing our results to Dirmeyer et al. (2009) Table 6 (i.e. "Top three external contributors of ES, expressed as a percentage of total precipitation over each nation.")

If this comparison is correct, the key countries, as well as corresponding percentages of contribution, match quite well, excepting for some differences in percent contributions to Bolivia. We will refer to this previous work given that it generally supports the results we find, while illustrating the disparity for volumes coming from Peru and Brazil to Bolivia.

RC6. P6 L6: Fig 5 is cited before Fig 4.

AC6. Thanks for the comment, and we have carefully checked all references to Figures and Tables to ensure that the references are (a) in the correct order, and (b)

that the Figures and Tables do not preceed their first reference.

RC7. P8 L23: What is "malnourished"? As written, the rangeland systems are. I think you mean the people in those systems. Likewise in L26-27, "areas" are not hungry, the people in them are.

AC7. Thank you, and indeed we did not write this correctly. We have corrected these, and carefully read the text for similar errors.

RC8. Fig 3: I think this must be mislabeled. The open circles must mark the in-country sinks and the colored dots are the various evaporation sources, no? Also, please expand or define the abbreviations "Pop." and "Resi." (actually it would be good to point out the threshold population density of 10 people/kmËĘ2 between the categories). Finally, what does the size of the circles indicate and what is the scale for that?

AC8. Thank you for the suggestions here. Based on your feedback Fig 3 has been completely remade (per your suggestion in Comment 9 below), and we no longer include the Anthrome data, nor the Anthrome colorbar. The land-use data is now presented as a histogram in a separate panel. A brief explanation of the Anthrome data, including that it incorporates a population density component, is found at the explanation of variables in section 2.4.

RC9. Secs 3.2.1 - 3.3.3 relative to Fig 3: It is not always evident by eye the assertions made regarding the relationships between elements in Fig 3. I think it would help to plot in each panel the first two moments (mean and standard deviation) as two

crossed whiskers (along X and Y axes): one for all the source areas (weighted by contribution - is that the size of the circles?) and one for the in-country sink. Then their differences and the overlap of the ranges of standard deviations can be easily seen, and statements like P10 L3-4 and L9-10 would have a better basis.

AC9. Thanks for the comment. This is an excellent suggestion, and indeed adds to both the clarity of the results, as well as provides more robust information about differences among sources and sinks. We have made this change to the figure, and can be seen in each of the case studies.

RC10. Fig 4: The color bar is very unclear. Log scale? The numbers are linear, and appear to be multiplied by 10Ȩ̈4; clearly not what the authors intend and not commensurate with the ranges in Verburg et al. (2011).

AC10. Thank you for the comment, and we agree that the colorbar was incorrect. In the interest of streamlining the text, we have since deleted this Fig 4 from the text.

RC11. Also Fig 4: In fact, I cannot see how Bolivia has such a high Market Influence index (so blue) based on the data of Verburg et al. (2011); their Fig 3 shows this to be very low for Bolivia. The colors seem to have more to do with the "qualitative" descriptions in Sec 3.4 than the quantitative data.

AC11. Thank you for the comment, and we agree that the colorbar was incorrect. In the interest of streamlining the text, we have since deleted Fig 4 from the text.

RC12. Also Fig 4: It would be good to note somewhere that the X-axis naturally correlates with the size of the country (small=low) and its aridity (dry=low), while the Y-axis correlates with continentality of the climate.

AC12. Thanks for the comment, as stated, we have deleted this figure.

RC13. P11 L29: "detectable changes in vegetation and associated changes in nearsurface meteorology" - please provide a reference for this statement.

AC13. Thank you, and the relevant citation was mistakenly included in the sentence immediately prior. This has been fixed now.

RC14. P12 L33-34: Not a sentence; appears to be missing a clause.

AC14. Thanks for the comment, and we have edited the sentence for clarity. It now reads:
"This is relevant primarily because significant areas of land acquired for agriculture (estimated at 360,000 hectares in GRAIN (2012)), could lead to extensive potential modification of the land surface, with associated impacts on moisture recycling."

RC15. Sec 3.4.2: The Niger case is missing a discussion of the economic links among neighboring countries like exists for the other cases. This makes the final paragraph much more "hand-wavy" than the discussions of the other two cases, in my opinion.

AC15. Thanks for the comment, and we have corrected this in the updated text.

Please see the new section on the economic interlinkages among the region and beyond.

RC16. Sec 3.4.3: Much is made about the strength of national land use regulations, but (1) in the particular case of Brazil they are highly variable in time, depending on which party is in power; (2) enforcement lags behind (this is discussed somewhat) and (3) the spatial and population scale of the problem makes such statements about regulation almost meaningless. The Acre region of Brazil, noted as a main external moisture source for the Bolivian precipitationshed, has experienced significant deforestation over the last 40 years, albeit not as widespread as Rondonia, which also borders Bolivia. The problem seems to be soft-peddled a bit here.

AC16. Thanks for the detailed attention to this section. We agree with the Reviewer here, and have modified this text considerably. We particularly draw attention to the fact that there is in the cases of the interior of the Amazon considerable discrepancy between the stated aims of government policy and the actual impact on the ground.

RC17. P14 L11-12: Likewise, the notion that the "region's land use is relatively wellgoverned with many controls in place to avoid large-scale change" seems untrue, and in contrast to the very next sentence. It is clear even from Google Maps that there is a very clear demarkation following the Bolivian border where deforestation is rampant in neighboring Brazil right up to the border.

AC17. The satellite imagery you point to on Google Maps is indeed compelling, and a literature review of relevant deforestation trends in this region bears out your observation. We have modified this section considerably to reflect this updated

information. We hope the updated text on deforestation policy, enforcement, and reality is now more consistent with the Reviewer's understanding of these systems.

RC18. Sec 3.5: In addition to explaining the archetypes and how they fit the previously presented data and social dynamics review, it should be frankly stated where they are unclear, or at odds.

AC18. Excellent point. We have included new Discussion section 4.1 "MRSES archetypes are idealized" where we explore these issues.

RC19. Fig 5: I also find this diagram somewhat unsatisfying, but perhaps I am not understanding it. Are the boxes meant to be static, or is it the fluctuations (changes) in the boxes that precipitate (pardon the pun) effects in other boxes by the arrows? For instance, Mongolia has a huge non-local (regional) evaporation source, small (9

AC19. Thanks for the comment, and for the clearly deep consideration of this figure. The archetype classification is subject to interpretation, which is what we did based on the synthesis of the different aspects of our analysis. Other reviewers echoed this feedback about the issue of subjectivity, and we have added several sentences at throughout the text that emphasize the blending of methods, and the explicit use of of an interpretive and subjective set of methods.

Furthermore, we include a new discussion section titled "MRSES archetypes are idealized" to discuss (a) where why the MRSES might be unclear or at odds with one another, (b) emphasizing that there are benefits and disadvantages to the different MRSES (per this Reviewer's suggestion), (c) emphasize that understanding MRSES requires subjective interpretation of results, given the blending of analytical approaches and the presence of value-based judgements (e.g. higher child malnutrition

is subjectively unfavorable).

RC20. P15 L6-7: "...large contributions..." of what? Be clear and complete. "...social processes driving the evaporation..." - this statement may be endemic of the communication problem. Presumably this is shorthand for "policy and economics drive land use change that affects evaporation potentially affecting downstream precipitation" but I think the authors forget how much better they understand their own material than their readers will.

AC20. We appreciate the feedback, and recognize that this ambiguous language can lead to confusion and frustration. We have modified the text in this specific location, and have considered this feedback throughout the entire paper.

RC21. P16 L5-6: This needs to be stated earlier, to clarify much of what goes before.

AC21. We agree, and have moved this to the beginning of the explanation of the archetypes indicating that since it is contained within all the archetypes it ought to be stated clearly at the beginning.

RC22. P16 L33-34: It is sentences like this that lead to confusion; the "regional factors that can influence...." means land use changes affecting moisture sources, right? If so, just say that. I think the hydrologic underpinnings get lost at times in this manuscript.

AC22. Thank you again, and we have fixed this language.

RC23. P17 L8: I think this is a typo: "and Brazil" should be "from Brazil"

AC23. Thanks, fixed.

RC24. Sec 3.5.3: Aren't the actual drivers of land use change (deforestation in the Amazon) much more from developed nations than is the case in Niger? Doesn't this also have implications for "tele-coupling", or not as defined?

AC24. Yes, and this was intended to be part of our point. Evidently we did not make the point well, so we have clarified the text.

RC25. P17 L28-30: This sentence would benefit from a concrete example or pointer to the specific data presented earlier.

AC25. Thank you for the comment, and we have provided a concrete example referring back to the isolated archetype Mongolia, in the form of Mongolian land-use policy.

RC26. P18 L1-2: Similarly, this sentence would benefit from an actual example of reinforcement/surprise, and not merely describing the situation.

AC26. Agreed. We have added an example of this in the context of Amazonian policy within Bolivia

RC27. Sec 3.8: This section I found to be more clearly presented. I am reminded of the interesting case evident in Wei et al. (2013; http://dx.doi.org/10.1175/JHM-D-12-079.1) where evaporation from irrigation in Northeast China appears to supply a significant amount of rainfall to North Korea!

AC27. Thanks for the comment, and this is an excellent example. We have added this in the second paragraph of this section.

RC28. P19 L9: "...gives Brazil power over Bolivia in potentially significant ways." I would phrase it that it gives Brazil "responsibility to Bolivia" - this is the other side of the coin from air pollution (including nuclear fallout), where it is pretty easy to track sources to those affected downstream. We are not used to thinking of water vapor in that way, but "responsibility" gives a more overarching concept to such linkages.

AC28. Thank you for the suggestion. We have thought about this phrasing, and have switched it to the following: "For example, demonstrating that Brazil is very important for Bolivia's rainfall potentially adds a matter for negotiation between the two countries, with all that entails, especially in terms of responsibility and power."

—————————————————————

---

## Author Comment (AC3) · 7 Mar 2018

Dear Editor,

We appreciate the opportunity to revise the paper, and respond to the Reviewers. The updated manuscript is much improved over the Discussion paper, and we are grateful for the time the Reviewers dedicated to carefully considering the content, and for providing guidance on how to improve the analysis and the writing.

We look forward to the Editorial decision.

Warmest regards,

Patrick Keys and Lan Wang-Erlandsson

---

## Author Comment (AC4) · 7 Mar 2018

Reviewer Comment = RC Author Comment = AC

RC Overview: The paper by Keys et al presents three case studies of links between the social and terrestrial moisture recycling system. This study combines quantitative modeling of terrestrial moisture recycling with metrics and a literature review of social factors. In this way, the study estimates the major sources of precipitation (i.e. precipitationsheds) for three case study countries. Gridded social variables are then evaluated for the source and sink nodes in each case study. Finally, a literature review is performed to reveal additional context for each case study and enable the development of moisture recycling social ecological systems archetypes.

[Figure]

Overall, I think this is an innovative, well-executed, and (reasonably) well-written paper that would make a unique contribution to the literature. I recommend publication after consideration of my comments below.

AC. We are pleased that the Reviewer has considered this work carefully. We hope the responses below address the Reviewers' concerns.
- - -
Major comments:
RC1. Human well-being/welfare has a precise definition in the social sciences literature. The term(s) "well-being" and "welfare" are used several times in the paper. I don't think these is the best term to use, since they mean something precise in the economics literature that is distinct to the meaning here. I think it would be better to refer to "social" aspects/variables/indicators of source and sink nodes. Then, the precise metric references should be specified whenever possible.

AC1. Thank you for the comment. We note that the phrase human well-being and welfare have specific definitions. As suggested by the reviewer, we use the specific phrases that describe the variable /indicator/ etc. and we make clear when using a general phrase that it is not already laden with pre-existing meaning.

RC2. Why not perform a global scale analysis? The literature review would be too difficult to perform for all countries in the world. However, a global scale analysis of precipitation-sheds and receiving countries would be relatively straightforward to perform. It appears the authors have all the information they need for this. They have WAM-2 pixels, social variables at the pixel scale. So, couldn't this be a global scale analysis for most aspects? Then, the 3 case study countries could be used for the literature review portion of the paper. If a global scale analysis is performed, then the authors will have more data to run some interesting regressions. For example, they

can calculate "precipitation-sheds" and "sink" nodes for all countries. Then, they can obtain average values of social variables in each source/sink. In this way, they will have enough statistical power to run multivariate regressions of the driving factors of the terrestrial moisture recycling system.

AC2. Thank you for the suggestion, but this level of analysis is beyond the scope of this present work of understanding the social aspects of a moisture recycling system. In terms of practical usefulness for stakeholders and the construction of MRSES, we think it is necessary to go deeper in understanding each moisture recycling system. Running the WAM-2layers for all countries globally, would be a considerable computational undertaking that, while compelling, addresses a somewhat different research question and target audience than at present. We also removed Fig 4 in order to streamline the paper. Thus, we suggest that a global scale analysis could be set aside for future work.

RC3. Fig 3 is confusing and could be simplified. There is a lot of information in Fig 3. I don't think most of it is necessary. For example, does the biome information convey anything interesting? There does not appear to be any trend between malnourished children (y-axis) and GDP/capita (x-axis), so this information could be made easier to read. I think this figure would be better if it presented the average value of malnourishment and GDP/capita explicitly for the source and sink region of each country. This might be able to be accomplished with a simple bar graph or box-whisker plot for each variable for each source/sink node. A table might even best illustrate upstream/downstream differences. This simplicity would better illustrate the main points made in sections 3.2.1-3.2.3.

AC3. We appreciate this suggestion for a revised Fig 3, and much of this is consistent with Reviewer 2. We have updated this figure to be a much simpler plot
showing sources and sinks, with corresponding mean and standard deviation. Also, the figures have been re-combined so that each case studies figures appear all together. This hopefully simplifies the information and assists interpretation.

RC4. The section on power dynamics could be improved. There seem to be many similarities between upstream/downstream power dynamics in precipitation-sheds and watersheds. I think this section would benefit from drawing from the power dynamics concepts in the transboundary watershed literature. A lot of work has been done on power/politics in international river asins that section 3.8 would benefit from referencing. Generally, section 3.8 could use a bit of a rewrite for clarity. Have any papers quantified the impact of upstream precipitation-sheds on downstream droughts? This seems like it would be the most clear example of upstream-downstream conflict/power issues. Also, can you expand on the Daw et al (2011) Reference? Does this paper specifically focus on power dynamics in teleconnected systems?

AC4. Thank you for this comment, and this is a very interesting suggestion. We have added several new sentences reflecting on the upstream/downstream power dynamics in watersheds, and how they are potentially similar or different to pre-cipitationsheds. The Daw paper focuses specifically on how there are trade-offs in ecosystem service benefits, and that a winner is often associated with a loser elsewhere. However, this is tangential to the core message of the research, and so we have removed this citation.

RC5. A bit more connection with the SES and socio-hydrology literature would be helpful. How does this work relate to socio-ecological systems (SES) work? Have similar archetypes (Fig 5) been presented in SES literature? Or socio-hydrology? What outcome variables are primarily of interest in the SES literature?

AC5. This is a great suggestion, and we have looked into similar discussions in the SES and socio-hydrology literature. The new section 4.3 "Advancing human-water systems understanding" and corresponding figure are included in the revised manuscript to address these considerations.

Minor comments:
RC6. The term "average market influence" is not clear and confusing. Please just call it what it is, i.e. GDP per capita.

AC6. The variable "average market influence" is actually a specific variable calculated in Verburg et al., (2011), that is a combination of (a) access to markets (calculated using proxies transport infrastructure, travel distance, and travel costs to major cities), and (b) per capita GDP. Thus, GDP per capita is not actually what the value is. However, other Reviewers have also pointed out that this variable is unclear, so we added a table that (among other things) provides clearer definitions of all variables throughout the text.

RC7. . Figure 4 doesn't seem to show much. What happens if you just plot national international moisture recycling (y-axis) against GDP per capita (x-axis)?

AC7. Thank you for the comment, and this feedback echoes the concerns of other reviewers. However, Fig 4 was considered no longer useful and has been removed from the manuscript.

RC8. P 18 line 14: "Though the analysis of environmental justice flows has

been simplified (Fig 3). . ..". Environmental justice flows are not quantified or presented in Fig 3. This statement is not warranted.

AC8. We agree with the Reviewer, and have removed the language of environmental justice, and refer more plainly to the specific variables we examine.

RC9. P 19 line 5: This sentence is a bit ironic. It seems to be a call for interdisciplinary scientists to engage and communicate with one another. However, this sentence is laced with jargon that is not widely understood (i.e. "positivism", "normative terminology")

AC9. This is ironic. We have defined these terms clearly now. Likewise, this feedback is generally consistent with other Reviewers, so we have read the text carefully and replaced or defined any remaining jargon.

---

## Author Comment (AC5) · 7 Mar 2018

Reviewer Comment = RC Author Comment = AC

**RC** Major comments**

The authors investigated terrestrial moisture recycling in three inland countries, namely, Mongolia, Niger, and Bolivia, by focusing on land-use change in moisturesource regions. By investigating land-use change policy of the countries in question and surrounding countries, the authors tried to explore the social dynamics of moisture recycling. Although I found the attempt quite interesting and novel, the manuscript in present form lacks clarity and quantitative evaluations in many parts. Hope the comments below are useful for further improvements.

AC. We appreciate the time that the Reviewer took to respond to the manuscript. We have made many improvements to the manuscript in response to all of the Reviewers, and hope that the updated manuscript meets the expectations of this Reviewer.

**RC** Specific comments**

RC1. Page 1 line 8 "We find that the sources and sinks of moisture can experience very different levels of human well-being, suggesting that power discontinuities must be included in the description of MRSES dynamics": How moisture "can experience different levels of human well-being"? What are "power discontinuities"?

AC1. Thank you for pointing out these issues, which were indicative of broader problems related to jargon and lack of clarity in our text. We have made major changes throughout the text to define terms and remove jargon.

RC2. Page 1 line 11 "This exploration of the social dimensions of moisture recycling": It seems an important precondition of this work that the "social dimension" plays an important role in terrestrial moisture recycling, but this is hardly proved (quantified) in text. I suppose the direct impacts of land-use change on the terrestrial hydrological cycle would be marginal. Exceptions are the cases for quite intensive irrigation (e.g. DeAngelis et al. 2010; Puma and Cook, 2010) and land-use change at continental and century-scale (e.g. Takata et al. 2009).

AC2. Thank you for this comment. In the original Introduction we stated "That the land-surface can and does influence the atmosphere is well-known (Dirmeyer and Brubaker, 1999; Domenguez et al., 2006; Tuinenburg et al., 2011; Bagley et al., 2012; Keys et al., 2016)."
It is apparent from the Reviewer's comment, however, that this sentence is insufficient to provide a convincing basis for proceeding toward the discussion of how social drivers of land-use change are a reasonable topic of discussion. As such, we have added a brief paragraph to clarify how land-use change modifies atmospheric moisture recycling, and then clearly state that a land-use change analysis is not the point of the present study, and that the interested reader should seek the original works that explore this topic. This updated text appears in the Introduction

"That land-use change can and does influence the atmospheric water cycle is wellsupported (e.g. Lo and Famiglietti, 2013; Wei et al., 2013; Halder et al., 2016; de Vrese et al., 2016). Impacts can include modifications of the energy budget (e.g. Swann et al., 2015), impacts to local or regional circulation (e.g. Tuinenburg et al., 2013), and impacts to the atmospheric water cycle (e.g. Spracklen et al., 2015; Badger and Dirmeyer, 2015)."

RC3. Page 4 line 3 "2.2 Tracking the sources of moisture": The authors applied the WAM2layers model to estimate the evaporation and precipitation of their study domain. First, I would suggest providing more detailed information on the boundary condition (i.e. simulation period, land-use assumption, validation data). Second, I would suggest conducting some additional simulations under counterfactual land-use which implies historical land-use change mentioned in Section 3.4. Such simulations would be highly effective to convince readers how significantly "social dynamics" would change precipitation or evaporation.

AC3. We appreciate the reviewers comments regarding the WAM-2layers and additional simulations. Evidently, our explanation of the WAM-2layers was insufficient, since the WAM-2layers does not in fact simulate anything. Rather, it is a moisture tracking scheme, that keeps track of the atmospheric water budget. In section 2.3, we explain how the WAM-2layers tracks moisture, and that we employ the ERA-Interim
Reanalysis data. Given that this Reviewer, and Reviewer 1, both thought we were simulating something, we have improved and clarified our explanation of the WAM-2layers to ensure there is no confusion of what the WAM-2layers does and does not do. This text may be found in section 2.3:

"We emphasize that the WAM-2layers is a moisture tracking scheme, and not a simulation. It is possible to couple the WAM-2layers with dynamic simulations of land-surface hydrology, including vegetation (e.g. Wang-Erlandsson et al., 2014; Keys et al., 2016), but that is not what we have done in this research. Thus, the results that we present are purely based on the implicit hydrological information contained within the ERA-Interim Reanalysis data."

While it would be very interesting to conduct counterfactual land-use changes to explore how various land covers influence the moisture recycling patterns, this is outside the scope of what we are writing about in this research.

Additionally, we recognize that the Reviewer would like to see additional evidence for how land-use change can significantly change precipitation and evaporation. However, we highlight (as noted in the previous comment) that the evidence for land-use change impacts on moisture recycling are well-established, and are thus not necessary to establish in this paper.

RC4. Page 8 line 26 "in general evaporation arising from relatively wealthier, less hungry areas falling out as precipitation in poorer hungrier areas": This part sounds very subjective. Add figures and tables to make this part quantitative and concrete.

AC4. Thank you for the comment, and we agree that this is unclear. Throughout Section 3.2 (in the original manuscript), we are referring to the results in Fig 3. However, this comment along with the feedback from the other Reviewers, suggests that we need to improve both the clarity of the text of Section 3.2 as well as the clarity
of Fig 3.

So, we have significantly modified Fig 3 to be much simpler and convey the information of evaporation source and precipitation sink characteristics much more robustly and clearly. Also, given that we now present each case study in its entirety, Fig 3 has been divided among the three case studies.

RC5. Page 10 line 3 "However there is a flow of moisture from wealthier areas to poor areas (relative)": Same comment as above.

AC5. Thank you for the comment, and we agree. Please see above comment for full details of the changes we made.

RC6. Page 10 line 7 "Within Bolivia itself, there is a cluster of wealthier rangelands and populated woodlands, and a cluster of much poorer remote and wild forest systems" Same comment as above. What is a cluster?

AC6. Thanks, and please see the comment above. We used cluster to refer to general grouping of the circles in the figure. However, since we have replaced Fig 3, with a much simpler figure, we have removed all text that refers to clusters (or the distribution of circles generally).

RC7. Page 10 line 8 "Surprisingly": Explain what is surprising. The authors tend to connect factor and factor subjectively. What are the solid knowledge based on established evidence here? In what sense surprising?

AC7. Excellent point, and we have removed this text. We agree that we should
not be injecting phrases like 'surprisingly' or 'interestingly' into the text.
RC8. Page 11 line 10 "affect moisture recycling policy": What is moisture recycling policy? In my view, the impact on moisture recycling is one of many (often unintended) secondary-impacts of land-use/industrial policy.

AC8. Moisture recycling policy (as far as I know) does not yet exist. The Reviewer is correct that changes to moisture recycling are going to be secondary impacts of land-use/industrial policy. Some work has been done to identify potential policies for direct governance of moisture recycling as well as for integrating moisture recycling into existing policy (e.g. Keys et al., 2017). However, no policies exist yet. We removed this sentence since it was confusing and unnecessary.

RC9. Page 14 line 29 "Construction of archetypes": Although it is an interesting idea that inland moisture recycling could be subdivided into three categories, I'm wondering how to find thresholds among them. Any region is neither fully isolated nor fully teleconnected. What to do with regions in between?

AC9. This is an excellent point and we agree with the Reviewer. Undoubtedly some regions may fall in between these archetypes, if not manifesting additional (as yet uncharacterized) dynamics that may yield entirely new archetypes.

Nonetheless, in the updated Section 3.4 and Section 4.2, we discuss the process of classification and how MRSES are likely to move from isolate toward regional toward tele-coupled, and that once they have become tele-coupled they are unlikely to reverse that trajectory.

RC10. Page 18 line 15 "in isolated systems (e.g. Mongolia) there can still be a wide range of well-being (e.g. wide range in poverty and malnutrition)": I couldn't follow the authors' logic. In every isolated systems the authors' claim holds true? Which figures/tables/sub-sections clearly do clearly support this claim?

AC10. This is a good point, and we have removed this text since it is unclear.

RC11. Figure 3: Very hard to understand. What does each plot represent (grid cells of each nation or those for each precipitationshed)? Also clearly indicate in text what we should focus on. These panels look random scatter without meaningful information at first glance.

AC11. Thank you, and your comment echoes the comments from all the Reviewers. We have re-made Fig 3 to be much clearer and communicate the intended information more simply. Likewise, given the restructuring of the cases (based on Reviewer 2 feedback), we present each panel of Fig 3 with its corresponding case study.

**References**

DeAngelis, A., Dominguez, F., Fan, Y., Robock, A., Kustu, M. D., and Robinson, D.: Evidence of enhanced precipitation due to irrigation over the Great Plains of the United States, J. Geophys. Res., 115, D15115, 2010.

Puma, M. J., and Cook, B. I.: Effects of irrigation on global climate during the 20th century, J. Geophys. Res., 115, D16120, 2010.

Takata, K., Saito, K., and Yasunari, T.: Changes in the Asian monsoon climate during 1700-1850 induced by preindustrial cultivation, P. Natl. Acad. Sci. USA, 106,
**ESDD**

---

## Author Comment (AC7) · 8 Mar 2018

Dear Reviewer 2,

Please see the Attachment under Reviewer #1, that links to a draft of the revised manuscript, which contains the changes detailed in the "Response to Reviewer". I recognize that this is not the typical procedure for Discussion papers, but the changes to the manuscript were so numerous among the four Reviewers that including this revised draft provides the clearest method for communicating our changes.

Regards, Patrick Keys

———————————————————

2017.

---

## Editor Comment (EC2) · M. Sivapalan (Editor) · 11 Mar 2018

The authors have been well served by four critical but constructive reviews. It seems from the review comments that, at the minimum, the presentation of the present needs substantial improvement. The authors are in agreement and so I would like a substantially revised manuscript to be submitted so I can get the revised manuscript revised one more time by (hopefully) most of the previous reviewers.

However, I would like the authors to also pay attention to the main contributions of the paper. What has been learned from this that we did not know before, why is this research significant etc? In particular, in the abstract the authors state that this "…. is part of an extension of the emerging discipline of socio-hydrology". In what way is

this an extension of socio-hydrology? The literature review of socio-hydrology is now substantial, although this is the first time there is work on focusing on atmospheric moisture recycling. Perhaps it is important to put this work in the context of what has been published before in socio-hydrology (I mean, if this is feasible). In spite of their claim, the literature cited is rather sketchy. Even if there is not an extensive literature review (there are other papers that do this already, e.g., Pande and Sivapalan in WIRES Water, Sivapalan and Bloeschl, WRR), it is important to draw a connection to previous work.

Apart from that, from my reading of the paper (including revised manuscript), I think the paper is very interesting, timely and eventually publishable, provided these comments can be satisfactorily addressed. Please submit a revised manuscript soon, so I can get it reviewed again promptly.

I look forward to receiving the revised manuscript soon.

---

## Referee Report (RR1)

**Review of 'On the social dynamics of moisture recycling' by Patrick W. Keys and Lan Wang-Erlandsson**

The article presents a new framework for understanding social ecological processes in regions sensitive to moisture recycling processes. The framework is novel and interesting. I applaud the authors for moving outside their comfort zone to give new insights on this issue. In general, however, I think the framework presented has limitations that I would like to see the authors critically reflect upon and explicitly mention in the text.

**General Comments**

**Two-way social feedbacks are only relevant in a minority of cases**

Biophysically it is clear that there is a one-way relation between source and sink regions regarding moisture recycling. I.e. changes in the source region can impact moisture recycling which can affect rainfall in the sink region. The authors state that social processes give rise to two-way feedbacks between the source and the sink. This is true but *only* in cases where there is tight coupling of trade in agricultural commodities between the source and sink region. For example, change in land use in the source region may reduce rainfall in the sink region. This reduces yields and increases market demand for agricultural crops. This demand can essentially be transmitted anywhere. There is only a positive socio-environmental feedback between the source and the sink if increased demand in the sink region (caused by reduced rainfall and reduced yield arising from land change in the source) is met by exports from the source region. I.e. increased demand in the sink may stimulate more land use change in the source which may reduce moisture recycling further. Other than this specific case, I cannot see a mechanism for socio-environmental feedbacks between the source and sink region. Please correct me if I am wrong on this.

Thus, to understand the relation between social changes and moisture recycling, you should focus on where the demand that drives land use change in the source region comes from. In most cases it is probably very local (subsistence), within a country or regional and international that you mention. You discuss these archetypes, but it is not obvious to the reader that they reveal that that two-way social feedbacks between source and sink are of relatively minor significance when you go to the regional or tele-coupled archetype. It seems that you are trying to fit an SES framework between source and sink regions with regard moisture recycling, when it will only be relevant in a minority of cases. SES within source and sink regions is no-doubt important, however.

**Comparison with river systems**

In general, the biophysical characteristics of a moisture recycling system are similar to that of a river system. I.e. upstream users impact downstream users via biophysical changes but not the other way around. However, social changes may be in both directions. A river system differs from the moisture recycling case as the leverage points, as you term them, are specific (such as governance of a reservoir), whereas in a moisture recycling scenario they are highly diffuse and heterogeneous. I.e. Large area of variable landscape types and agricultural practices. Equally, the social and economic links are similar but perhaps stronger in river system, owing to the organisation of societies and trade around rivers. Also, users clearly understand where source and sink regions are in a river system, whereas this is not widely known in moisture recycling systems, and thus issues such as fairness of resource sharing are not widely recognised. There has been much work done examining upstream and downstream social and biophysical feedbacks in river systems which would provide a good template for this work, rather than trying to reinvent the wheel. I would like to see a more

complete review of this literature (I only see the reference of Grumbine et al., 2012) and explanation of the similarities and differences between a river system and a moisture recycling system.

**Specific Comments**

Page 1 Line 18-19 "socio-meteorology and socio-climatology" These concepts are surely central to the IPCC working group 2 and 3 reports.

Page 2 Line 12: Why should it have a social focus? What if natural scientists are examining the biophysical processes? Perhaps you mean "social component"

Page 2 Line 20 unexplored. These

Page 3 line 1-10 Are you using an SES Framework? If so, outline specifically the framework you use, rather than SES type of thinking.

Page 3 line 11-12 "Hydrologists specifically", however moisture recycling is more in the subdiscipline of meteorology that focuses on land-atmosphere feedbacks.

Page 5 Tele-connection definition. I have never heard of a teleconnection referring to connection separated by time alone. In climatology I am only familiar with connections separated by space or space and time combined.

Page 7 line 16 km is in a different typeface.

Defensive language used which doesn't come across well: e.g. Page 7 line 30-34, Page 8 line 4-5, Page 23 Line 22 – 34

Page 8-19 In the results you cover many aspects of the economy within and between countries and many traded commodities. In one way every aspect of an economy is inter-related either directly or indirectly. However, the complexity of economies is such, that it seems necessary to narrow down analysis to the most relevant variables. Therefore, it would be helpful to constrain analyses to the agricultural economy. For example, France, Thailand, Malaysia and China are Niger's major trading partner (in terms of value) but not in agricultural products, thus what is the relevance here? You can find agricultural trade data from the FAOstat website.

Page 17 Line 13 - 25 It seems that these two paragraphs are only relevant to a source region. You should state here whether you refer to source, sink or the link between the two.

Page 18 Fig 5. I find this figure too complex. I would prefer an individual diagram for each archetype. Also, it is unclear what the arrows mean. Are they biophysical fluxes or social influence or resource flows? Some indication of the difference between material and immaterial links would be helpful as you have done in figure 1.

Page 20 Line 1. What feedbacks?

Page 21 Line 27: You talk about feedbacks but how do these actually emerge in your system? Feedbacks emerge from mechanisms that reinforce an initial change in the direction of that change. I guess here you are discussing positive socio-environmental feedbacks. So, what changes in the social or environmental realm (in source, sink or between the two) and how is that reinforced (in source, sink or between the two).

Page 23 line 10 pitfalls of their

Page 23 line 14 is a prerequisite

Page 23 Line 7-20 Lecturing of natural scientists seems to me slightly ungrounded. What is your evidence that nearly all natural scientists assume that every meaningful assertion ought to be verifiable or provable logically or mathematically? This is certainly the case when dealing with natural systems but in my experience there is openness of those engaging in interdisciplinary science to alternative worldviews. Either way, we natural scientists, and sound social scientists, certainly like evidence, however you provide none for these assumptions. Perhaps, perceptions of natural and social scientists on interdisciplinary science could be a study by itself grounded upon sound social science approaches such as surveys, co-publication network analysis etc. Thus, I would leave this part out of your paper or at least dampen the tone a great deal.

Page 23 line 33 well with research

---

## Author Response (AR2)

May 15, 2018

Dear Professor Sivapalan,

Thank you for the opportunity to revise and resubmit our manuscript "On the social dynamics of moisture recycling." We are pleased to hear that the revisions were well-received by the Referees and yourself. We have made the changes suggested by the Referees, and where we did not change, we explained our rationale. We indicate each original comment of a Referee with the prefix 'RC', and the Author Comment with 'AC'.

I hope that the revised manuscript meets your expectations for Earth System Dynamics, and I look forward to your decision.

Warm regards,
Patrick Keys on behalf of co-author

REFEREE #1 (ANONYMOUS)
RC1: I found that the authors have put substantial additional efforts and thoroughly revised the manuscript considering all comments provided. Key questions driving the study have been added, both the quantitative and qualitative aspects of the study have been clearly highlighted, and all other comments have been addressed. Overall, the revised manuscript reads very well. I have only a few minor comments for the authors.
AC1: Thank you for the response. We will address your comments below

RC2: Perhaps I overlooked this in the previous review or I missed something in the revised version, but I wonder if there was a basis for the use of 1mm for boundary demarcation. In the current version, it is noted that the authors followed Keys et al. (2014); further elaboration would be appreciated.
AC2: Thank you for pointing this out. The basis for the 1mm boundary in this instance was to have a 'physical' boundary that was meaningful, primarily in the sense that common rain gauges have a lower limit of 1mm. Thus, theoretically, a region that contributes 1mm of rainfall is an approximate lower limit for what might be detectable in terms of change on common rain gauges. We will highlight this more clearly in the text where we discuss the 1mm boundary, so that our rationale is clear.

RC3: At numerous instances, the authors discuss policy relevance. For example, the text around lines 22-25 in page 17 and lines 7-10 in page 18. It is still unclear to me about how moisture recycling has been used in policy making/implementation. In particular, the authors note that, in regions with rule-of-law, evaporation is used in government regulations and policies. Could the authors elaborate this further and/or provide some examples/references about where and how this is done?
AC3: Thank you for this comment. We were evidently unclear in the text regarding policy relevance. To the best of our knowledge moisture recycling has not been used in policy making/implementation. Previous work has hypothesized approaches to moisture recycling governance, but we are unaware of explicit moisture recycling policies. We will make this distinction more clear.

REFEREE #2 (Paul Dirmeyer)
RC: The revised manuscript is much improved, and much much easier for the reader to follow. The reorganization of the manuscript has made the message much clearer and better serves the intent of the authors. Table 1 greatly aids readers in understanding the terminology used - few if any readers will be well versed in all of these terms, as you are bringing together concepts from (previously) disparate lines of research. Likewise, the new version of Fig 1 is a big improvement. Also, I appreciate the added specific examples that are given throughout Section 4 now - very helpful for comprehension.

All-in-all the manuscript is now of high quality - I suggest only minor revisions (mostly technical but a couple more substantial) based on the specific comments below. All comments below reference pages and line numbers in the marked-up version of the manuscript that accompanied the responses to reviewers.

AC: We very much appreciate the positive feedback from the Referee. We have made the suggested changes that you highlight below, and where we did not we explain our rationale.

Specific comments:

RC1: P2 L16: The term "unpacking" is informal, colloquial jargon; "understanding" would be a better choice.
AC1: Agreed, we have made this change.

RC2: P2 L27: "unexploredThese" needs a period and space in between.
AC2: Thank you, this text has been removed.

RC3: P3 L5: change "have" to "has"
AC3: Thank you, this has been changed.

RC4: Table 1: Next to "Social-ecological systems" put the abbreviation "(SES)" as it is used in the text.
AC4: Thank you, we have added this.

RC5: P6 L3: delete "similar"
AC5: Thank you, this has been changed.

RC6: P9 L7: "grey literature" is also informal and a bit pejorative. Using "non peer-reviewed literature" may be a better choice.
AC6: Thank you for the comment. We were not aware this can be interpreted as pejorative. This has been changed.

RC7: Fig 2: This appears to be a glitch only with this figure as it does not appear in Figs 3 and 4, but there is a blue background to panel (b) that makes the labels unreadable.
AC7: This is only present in the 'tracked changes' document. Not in the revised LaTeX file. We have made certain that the figure is normal, and readable, in the final LaTeX and PDF files.

RC8: Fig 2: In panel (d), what does a negative value for "malnourished children" mean? The whisker goes to -20.

AC8: This is a mistake and should of course not go below 0. We have fixed this. We have also made sure that there are no other such errors in the other files.

RC9: Fig 2 caption: Change "between country and" to "between the country and its".

AC9: Thank you, this has been changed for all figure captions.

RC10: Figs 2-4: Panel 3 in each, it is hard to digest this information as presented. I think it would help to order the categories on the abscissa by their source values (blue bars), from largest to smallest.

AC10: Thank you, this is a great idea. We experimented with different ways to present this information. It is not perfect, since the sorting of anthromes for the source areas is not the same as the same as sorting the anthromes for the sink. However, I think the sorted anthrome graphs are somewhat easier to interpret.

RC11: P12 L3: "kids" is informal; say "children". Also, change "evaporation key sources" to "key sources of evaporation".

AC11: Agreed on both counts. This has been changed.

RC12: P13 L19-20: "associated changes in near-surface meteorology"; of what sort? Please give specifics here.

AC12: Thank you for this comment. We have more closely read the referenced article, and we mis-attributed their findings. They quantified changes in NDVI, and *hypothesized* changes and suggested implications for atmospheric processes. We've adjusted our statement accordingly.

RC13: P14 L9: Delete redundant "from".

AC13: Thank you, this has been changed.

RC14: Sec 3.2.3: The first 2 sentences are largely redundant with text above.

AC14: We agree, and this has been changed.

RC15: Fig 3: I may have said this before: the large Mediterranean source may be bogus. This is a problem in our estimates using QIBT also, which we were never able to eliminate - a general problem of after-the-fact approaches including WAM-2layers to estimating sources from reanalyses data at low time resolution (much lower than model time steps). The low-level convergence of moisture over the Sahel in the rainy season is between moist air coming from the south and very dry air from the north. The two sides get convolved and impossible to sort out from reanalysis wind fields when tracing backwards in time. Atmospheric models that include water vapor tracers do not show such large moisture transports from the Mediterranean across the Sahara.

AC15: Thank you for the comment, and we appreciate the nuanced comparison between our two approaches to moisture tracking. Based on your suggestion, we have adjusted the boundaries of our figure, and added some comments about this potentially spurious finding (see below). We did not want to arbitrarily remove this section from the actual analysis, but re-framing the plotted domain in Fig 3, so that the Mediterranean sources do not appear may help avoid any confusion.

We hope these changes are sufficient in your view.

"Our method detected some contribution from the Mediterranean Sea, but this is likely spurious, and due to the inability of moisture tracking models such as the WAM-2layers (as well as other *a posteriori* tracking models) to disentangle the origin of moisture convergence in this region, specifically the mixing of dry air from the north with moist air from the south."

RC16: P16 L7 (or actually in the reference): GRAIN is an acronym for "Genetic Resources Action INternational" that needs to be expanded.
AC16: Thanks you for this comment. We have actually found an alternative reference, since there are open-access sources on this particular topic, which may be more reliable than sources from advocacy organizations.
http://www.landmatrix.org/en/about/#how-should-i-cite-the-land-matrix-global-observatory

RC17: P16 L27: "influence" implies intent; better to say "affect".
AC17: Thank you for this comment, this has been changed.

RC18: P17 L7-8: change "to a low concentration of moisture supply covering Brazil's entire domain" to "to the remainder of Brazil supplying a low concentration of moisture from a large area"
AC18: Thank you for the suggestion, this has been changed.

RC19: P17 L14: "72%" must be a typo. Maybe that's 7.2%?
AC19: Indeed, this should be 7.2%, this has been changed. Thank you for catching such a glaring mistake.

RC20: P17 L30: Change "led" back to "lead" - needs to be present tense.
AC20: Thank you, we agree; this has been changed.

RC21: P19 L12: "Rondônia should have a circumflex over the second o.
AC21: Thank you, we have added the circumflex.

RC22: P21 L29: Change "on precipitation" to "of precipitation".
AC22: Agreed, this has been changed.

RC23: P22 L6: Change "contribution" to "contributions".
AC23: Agreed, this has been changed.

RC24: P22 L11: Add "For instance," before "Nigeria experiences..."
AC24: Thank you, this has been changed.

RC25: Sec 3.4.3: Note somewhere that this subsection refers to the green lines in Fig 5.
AC25: Great suggestion, we have added this to the paragraph.

RC26: P26 L24: I suggest changing "ability" to "potential" or "possibility" - as this is not a solidly proven result. To my knowledge, no one has done water isotope analysis, for instance, to establish the veracity of this model result.
AC26: Fair point, we have changed the language to more accurately reflect the statement.

RC27: Bottom P 27: Change "export of moisture" to "export of atmospheric moisture" so it is clear you are not referring to transboundary rivers.
AC27: Thank you, this has been changed.

REFEREE #5 (ANONYMOUS)
Review of 'On the social dynamics of moisture recycling' by Patrick W. Keys and Lan WangErlandsson

RC1: The article presents a new framework for understanding social ecological processes in regions sensitive to moisture recycling processes. The framework is novel and interesting. I applaud the authors for moving outside their comfort zone to give new insights on this issue. In general, however, I think the framework presented has limitations that I would like to see the authors critically reflect upon and explicitly mention in the text.

AC1: Thank you for the detailed and thoughtful comments.

General Comments
RC2: Two-way social feedbacks are only relevant in a minority of cases. Biophysically it is clear that there is a one-way relation between source and sink regions regarding moisture recycling. I.e. changes in the source region can impact moisture recycling which can affect rainfall in the sink region. The authors state that social processes give rise to two-way feedbacks between the source and the sink. This is true but only in cases where there is tight coupling of trade in agricultural commodities between the source and sink region. For example, change in land use in the source region may reduce rainfall in the sink region. This reduces yields and increases market demand for agricultural crops. This demand can essentially be transmitted anywhere. There is only a positive socio-environmental feedback between the source and the sink if increased demand in the sink region (caused by reduced rainfall and reduced yield arising from land change in the source) is met by exports from the source region. I.e. increased demand in the sink may stimulate more land use change in the source which may reduce moisture recycling further. Other than this specific case,I cannot see a mechanism for socio-environmental feedbacks between the source and sink region. Please correct me if I am wrong on this.
AC2: Thank you for this thoughtful comment. The primary way that we include trade in our discussion is via the key import and export partners discussion, the brief section on land acquisitions, and the inclusion of 'markets' in Figure 5 or the Archetypes discussion. We do certainly agree that trade is a critical aspect of sink-to-source connection, but we think there are several other kinds of connections. First, political power could be a connection, where a sink region has more political power than a source region and can thus mandate that certain land-use actions take place. Second, international law could be used (hypothetically) if a sink region can prove it experienced harm as a result of actions taken by a source region. Third, informal or nonbinding agreements could serve as a platform for sink and source regions to discuss land-use outside of regulatory action or political mandate. These are just a few of the examples that we discuss in the paper, but we think they are indicative of a much broader set of social connections than trade alone.

RC3: Thus, to understand the relation between social changes and moisture recycling, you should focus on where the demand that drives land use change in the source region comes from. In most cases it is probably very local (subsistence), within a country or regional and international that you mention.

AC3: We agree that trade can be a useful, but we do not think it ought to be the only focus for identifying relevant land-use changes.

RC4: You discuss these archetypes, but it is not obvious to the reader that they reveal that that two-way social feedbacks between source and sink are of relatively minor significance when you go to the regional or tele-coupled archetype. It seems that you are trying to fit an SES framework between source and sink regions with regard moisture recycling, when it will only be relevant in a minority of cases. SES within source and sink regions is no-doubt important, however.

AC4: Thank you for the comment. The SES framework as we use it simply a basis for thinking about human-environment interactions as a coupled system. We intentionally steer clear of more in-depth SES theory since we believe that is outside the scope of this particular work. We consider the SES framework to remain relevant at larger scales, in part because as the system expands, the source and sink regions will remain SES regardless of the regional and tele-coupled dynamics. The social connections, we argue, are what make the SES expand spatially, beyond the local scale. We will, however, reflect on this discussion for future work.

Comparison with river systems
RC5: In general, the biophysical characteristics of a moisture recycling system are similar to that of a river system. I.e. upstream users impact downstream users via biophysical changes but not the other way around. However, social changes may be in both directions. A river system differs from the moisture recycling case as the leverage points, as you term them, are specific (such as governance of a reservoir), whereas in a moisture recycling scenario they are highly diffuse and heterogeneous. I.e. Large area of variable landscape types and agricultural practices. Equally, the social and economic links are similar but perhaps stronger in river system, owing to the organisation of societies and trade around rivers. Also, users clearly understand where source and sink regions are in a river system, whereas this is not widely known in moisture recycling systems, and thus issues such as fairness of resource sharing are not widely recognised. There has been much work done examining upstream and downstream social and biophysical feedbacks in river systems which would provide a good template for this work, rather than trying to reinvent the wheel. I would like to see a more complete review of this literature (I only see the reference of Grumbine et al., 2012) and explanation of the similarities and differences between a river system and a moisture recycling system.

AC5: Thank you for this very insightful comment. We agree that there is a great deal to discuss regarding the many possible similarities and dissimilarities between moisture recycling systems and river systems. However, the amount of material that is required for this comparison and discussion is far too much to include in the current manuscript. We aimed for this manuscript to be an introduction of this idea of social connections within moisture recycling systems, and that hopefully it can serve as a springboard for other research to discuss, critique, or even include as a basis for future MRSES study. If we do not have an opportunity to tackle this important comparison between precipitationsheds and watersheds, we hope that someone else does.

Specific Comments
RC6: Page 1 Line 18-19 "socio-meteorology and socio-climatology" These concepts are surely central to the IPCC working group 2 and 3 reports.

AC: These are emerging disciplines with some nascent discourse related to weather, climate, and deeper integration of society and sociology within the research itself. There are a handful of references to either concept, but only ever in passing. Though there is of course a vast field of meteorological and climate impacts research, we do not think that is the same as a true blending of social science methods with meteorology or climatology. It is certainly true that not everything needs a new discipline or a new name, so it will be interesting to see to what extent these emerging notions of socio-meteorology and socio-climatology gain traction, or wither away.

RC7: Page 2 Line 12: Why should it have a social focus? What if natural scientists are examining the biophysical processes? Perhaps you mean "social component"
AC7: Thank you for the suggestion, we will make the change to component.

RC8: Page 3 line 1-10 Are you using an SES Framework? If so, outline specifically the framework you use, rather than SES type of thinking.
AC8: Thank you for the suggestion. Since we are using SES type of thinking we consider our treatment of the field sufficient for this work. We are drawing attention to the theoretical lineage of Holling, Gunderson, and Folke with our citations (later on Page 3), which for our purposes is illustrative of how we are using the concept. As we said previously, we are not delving into the theory of different SES approaches, and rather using the SES approach more broadly.

RC9: Page 3 line 11-12 "Hydrologists specifically", however moisture recycling is more in the subdiscipline of meteorology that focuses on land-atmosphere feedbacks.
AC9: This is a very good point. We will change the sentence to say: "will provide Earth system scientists who study the atmospheric water cycle"

RC10: Page 5 Tele-connection definition. I have never heard of a teleconnection referring to connection separated by time alone. In climatology I am only familiar with connections separated by space or space and time combined.
AC10: This is an excellent point, we have changed this to "separated by space or space and time."

RC11: Page 7 line 16 km is in a different typeface.
AC11: Thank you for catching this, we have fixed this.

RC12: Defensive language used which doesn't come across well: e.g. Page 7 line 30-34, Page 8 line 4-5, Page 23 Line 22 – 34
AC12: We agree this is coming off as defensive (and in some cases offensive) and will adjust the phrasing accordingly. Part of this work was to jounce the reader a bit out of the normal scientific delivery of material, but still, this is too much. Thank you for highlighting this.

RC13: Page 8-19 In the results you cover many aspects of the economy within and between countries and many traded commodities. In one way every aspect of an economy is inter-related either directly or indirectly. However, the complexity of economies is such, that it seems necessary to narrow down analysis to the most relevant variables. Therefore, it would be helpful to constrain analyses to the agricultural economy. For example, France, Thailand, Malaysia and

China are Niger's major trading partner (in terms of value) but not in agricultural products, thus what is the relevance here? You can find agricultural trade data from the FAOstat website.

AC13: Thank you for this comment. While we agree that agriculture is very important, it is not the only land-use change that matters. For example, large-scale changes in forest cover or extensive mining can have significant impacts to evaporation. We certainly agree that considering a full economy is too complex. However, our purpose for considering who the trading partners were was to identify whether trading partners overlapped with key moisture source regions. For example, in the case of Niger, its trading partners did not overlap very much with the precipitationshed, so the economy was not considered a very important social connection.

RC14: Page 17 Line 13 – 25 It seems that these two paragraphs are only relevant to a source region. You should state here whether you refer to source, sink or the link between the two.

AC14: Thank you. When reading this section with your comment in mind, we now see that it can be confusing. We will add a sentence clarifying our meaning.

RC15: Page 18 Fig 5. I find this figure too complex. I would prefer an individual diagram for each archetype. Also, it is unclear what the arrows mean. Are they biophysical fluxes or social influence or resource flows? Some indication of the difference between material and immaterial links would be helpful as you have done in figure 1.

AC15: Thank you for your comment. Earlier referee comments suggested that the figure would be better as a single figure, which we agree with, since it is possible to see the whole structure at once. We explored how to add more information to the figure, e.g. physical water flows vs. 'policy flows', but the figure rapidly became too cluttered. Your suggestion does highlight a need for a better statement of our explicit exclusion of this information for visual simplicity. We will add text explaining this.

RC16: Page 20 Line 1. What feedbacks?

AC16: Thank you. We have added an example of the type of feedbacks we implied.

RC17: Page 21 Line 27: You talk about feedbacks but how do these actually emerge in your system? Feedbacks emerge from mechanisms that reinforce an initial change in the direction of that change. I guess here you are discussing positive socio-environmental feedbacks. So, what changes in the social or environmental realm (in source, sink or between the two) and how is that reinforced (in source, sink or between the two).

AC17: Thank you for the comment, and we have added an example of this to the text, per the Refereee's previous comment.

RC18: Page 23 line 10 pitfalls of their

AC18: Thank you, this text was changed in response to RC20 below.

RC19: Page 23 line 14 is a prerequisite

AC19: We agree.

RC20: Page 23 Line 7-20 Lecturing of natural scientists seems to me slightly ungrounded. What is your evidence that nearly all natural scientists assume that every meaningful assertion ought to

be verifiable or provable logically or mathematically? This is certainly the case when dealing with natural systems but in my experience there is openness of those engaging in interdisciplinary science to alternative worldviews. Either way, we natural scientists, and sound social scientists, certainly like evidence, however you provide none for these assumptions. Perhaps, perceptions of natural and social scientists on interdisciplinary science could be a study by itself grounded upon sound social science approaches such as surveys, co-publication network analysis etc. Thus, I would leave this part out of your paper or at least dampen the tone a great deal.

AC20: Thank you for the thoughtful comment. We agree that the phrasing is too strong. We will dampen the tone.

RC21: Page 23 line 33 well with research

AC21: Thank you for catching this. We have changed it.